# Molecular insights into receptor binding energetics and neutralization of SARS-CoV-2 variants

Melanie Koehler [1,5], Ankita Ray [1,5], Rodrigo A. Moreira [2], Blinera Juniku[1], Adolfo B. Poma [3✉] &
David Alsteens [1,4✉]

Despite an unprecedented global gain in knowledge since the emergence of SARS-CoV-2, almost all mechanistic knowledge related to the molecular and cellular details of viral replication, pathology and virulence has been generated using early prototypic isolates of SARS-CoV-2. Here, using atomic force microscopy and molecular dynamics, we investigated how these mutations quantitatively affected the kinetic, thermodynamic and structural properties of RBD—ACE2 complex formation. We observed for several variants of concern a significant increase in the RBD—ACE2 complex stability. While the N501Y and E484Q mutations are particularly important for the greater stability, the N501Y mutation is unlikely to significantly affect antibody neutralization. This work provides unprecedented atomistic detail on the binding of SARS-CoV-2 variants and provides insight into the impact of viral mutations on infection-induced immunity.

[1] Louvain Institute of Biomolecular Science and Technology, Université catholique de Louvain, Louvain-la-Neuve, Belgium. [2] Institute of Fundamental Technological Research, Polish Academy of Sciences, Pawińskiego 5B, 02-106 Warsaw, Poland. [3] International Center for Research on Innovative Biobased Materials (ICRI-BioM)—International Research Agenda, Lodz University of Technology, Żeromskiego 116, 90-924 Lodz, Poland. [4] Walloon Excellence in Life sciences and Biotechnology (WELBIO), 1300 Wavre, Belgium. [5] These authors contributed equally: Melanie Koehler, Ankita Ray. ✉email: adolfo.poma-bernaola@p.lodz.pl; david.alsteens@uclouvain.be

One and half year since the start of the COVID-19 pandemic, we are just beginning to understand how the severe acute respiratory syndrome novel coronavirus (SARS-CoV-2) can overcome host cell entry barriers and outsmart the human immune response[1,2]. Fortunately, several vaccines have been developed that can currently prevent COVID-19 with high efficacy, and their roll out is expected to help control the pandemic[3]. However, with little more than 10% of the world population presently (July 2021) vaccinated, the continuous emergence of SARS-CoV-2 variants with reduced susceptibility to infection- and vaccine-induced immunity, and the vast animal reservoir of SARS-like viruses, there is no time for complacency[4]. Rapidly spreading SARS-CoV-2 variants of concern (VoCs) demonstrate how easily this virus can accommodate antigenic changes in its spike (S) protein without any loss of fitness[5].

Despite an unprecedented global gain in knowledge since the surge of SARS-CoV-2, almost all mechanistic insights related to molecular and cellular details of viral replication, pathology and virulence have been generated using prototypic early SARS-CoV-2 isolates (such as the original Wuhan isolate). Unexpectedly, however, SARS-CoV-2 evolved into a moving target with VoC, such as the infamous Indian variants (Kappa and Delta), and Variants of Interest collectively now representing over 90% of GISAID sequence entries[6]. As for all viruses, entry is a key step in the SARS-CoV-2 life cycle. In this regard, some mutations in SARS-CoV-2 are responsible for large conformational changes that may enhance infection, such as those located in the distal region of the spike protein and near the fusion region as the well-known D614G. They have been shown to facilitate transitions from closed to open state prior to ACE2 binding[7–9]. Others placed in the receptor-binding motif (RBM) enhance recognition with ACE2 receptor and do affect the efficiency of monoclonal antibody (mAb) neutralization. The RBM, which is part of the receptor-binding domain (RBD) localized at the top of the trimeric SARS-CoV-2 spike (Fig. 1a, b), is a prime target for neutralizing antibodies that are induced by infection or by currently used COVID-19 vaccines[10]. The RBM is responsible for binding angiotensin-converting enzyme 2 (ACE2) as a primary receptor for the virus (Fig. 1b). The RBM, however, is very tolerant to mutations (Fig. 1c), many of which do not affect or even enhance human ACE2 receptor binding but may be detrimental for antibody recognition. N501Y, for example, present in the Alpha, Beta, and Gamma variants, increases ACE2 binding[11,12]. One of the far-reaching consequences of this antigenic variation of the RBM was the withdrawal of Eli Lilly's mAb bamlanivimab from emergency use approval as monotherapy for the treatment of COVID-19 patients, because VoCs with E484K/Q mutations had become resistant. Current knowledge, despite all the enormous efforts, threatens to be outdated if not anticipating on future evolutions, therefore a better understanding on the impact of spike protein mutations on the spike-receptor interaction at atomic resolution is of pivotal importance as well as its influence on its inhibition by antibodies.

Previously, we have reported the use of atomic force microscopy (AFM) to map the interaction forces between the AFM tips functionalized with the S-glycoprotein of RBD wildtype (WT) and the ACE2 receptors on model surfaces. By single-molecule force spectroscopy (SMFS) and biolayer interferometry (BLI), we have shown that there is a high affinity (~120 nM) guided by specific multivalent interactions at the RBD—ACE2 interface[13]. In this study, by using force—distance (FD) curve-based AFM, we have derived the kinetic and thermodynamic parameters between the ACE2 receptors on model surface with RBD of four different SARS-CoV-2 mutants (Alpha, Beta, Gamma, and Kappa respectively) (Fig. 1c–e). In addition to single-molecule force spectroscopy, we also elucidated the molecular basis of increased transmissibility. In this regard, all-atom molecular dynamics (MD) simulation has been employed to characterize the energetic and structural changes for the RBD—ACE2 complex interface. Our energy estimation captures the local contribution of the non-bonded energies for a single residue with respect to the whole ACE2 and hence it correlates with the appearance or disruption of residue−residue contacts. In this regard, the energy scale of the RBD—ACE2 interface should not be confused with the thermodynamic binding free energy. Thus, the monitoring of Lennard−Jones and electrostatic energies and the strength of residue contacts established at the interface provide a good description of local and long-range effects caused by mutations. Alpha, Beta, Gamma, and Kappa VoCs share common mutations which affect the close environment of the mutated amino acids. Furthermore, some residue contacts far away from those amino acids vary towards high strengths (or frequencies in MD) which induce an effect of structural stabilization, in particular for Gamma and Kappa mutants. Finally, we tested the neutralization efficiency of two mAbs, obtained by immunization of a mouse with the WT-RBD. Strikingly, we observed that one mAb shows excellent anti-binding properties against all variants while the second tested one lost its neutralization potential for the three variants having the E484 mutation.

## Results

**VoCs RBD binding to ACE2 is facilitated**. Several mechanisms might account for increased variant transmissibility, such as increased spike protein density, greater furin cleavage accessibility, and enhanced spike protein binding affinity for the ACE2 receptor[14,15]. To test whether the VoCs bind ACE2 with an increased affinity, binding of purified Alpha-, Beta-, Gamma- and Kappa-RBD were compared with binding of the WT-RBD, using FD-curve based AFM. To probe RBD-ACE2 complexes in vitro, monomeric ACE2 receptors were covalently immobilized via NHS/EDC chemistry on OH-/COOH-terminated alkanethiol grafted gold-coated surfaces (see "Methods"). The various VoC RBD proteins were covalently attached to the free end of a long, polyethylene glycol (PEG)$_{24}$ linker linked to the AFM tip[13,16] (Fig. 1d). To investigate the binding potential, the RBD modified tips were cyclically approached and retracted from the ACE2 surface. The force acting between the functionalized tip and the surface, which isexpressed in piconewtons (pN), was monitored over time, resulting in force vs. time curves (Fig. 1e)[16,17]. Upon retraction, binding events were commonly observed at rupture distances >12 nm (corresponding to the full extension of the PEG linker and used as a signature of specific unbinding events), with binding frequencies as follows: Alpha-RBD ≈ Beta-RBD ≈ Gamma-RBD ≈ Kappa-RBD > WT-RBD (Fig. 2a). The specificity of these interactions was confirmed by control experiments using (i) an AFM tip only functionalized with the PEG linker or (ii) toward NHS/EDC surfaces missing the receptor. The dashed boxes in Fig. 2a illustrate a significant decrease in adhesion when the RBD strain was absent or when the interaction was studied between the RBD functionalized AFM tip and receptor lacking surface, confirming that the vast majority of the recorded binding events stemmed from specific interactions. The binding frequencies are in good agreement with the data previously reported for the WT-RBD using the same approach[13]. For the VoCs, the higher binding frequencies observed suggest that their binding to ACE2 are facilitated in comparison to the WT, which is in good agreement with the higher transmissibility observed of those VoCs.

**Thermodynamics of VoCs-RBD—ACE2**. Having shown that VoCs have a higher binding potential, we then investigated

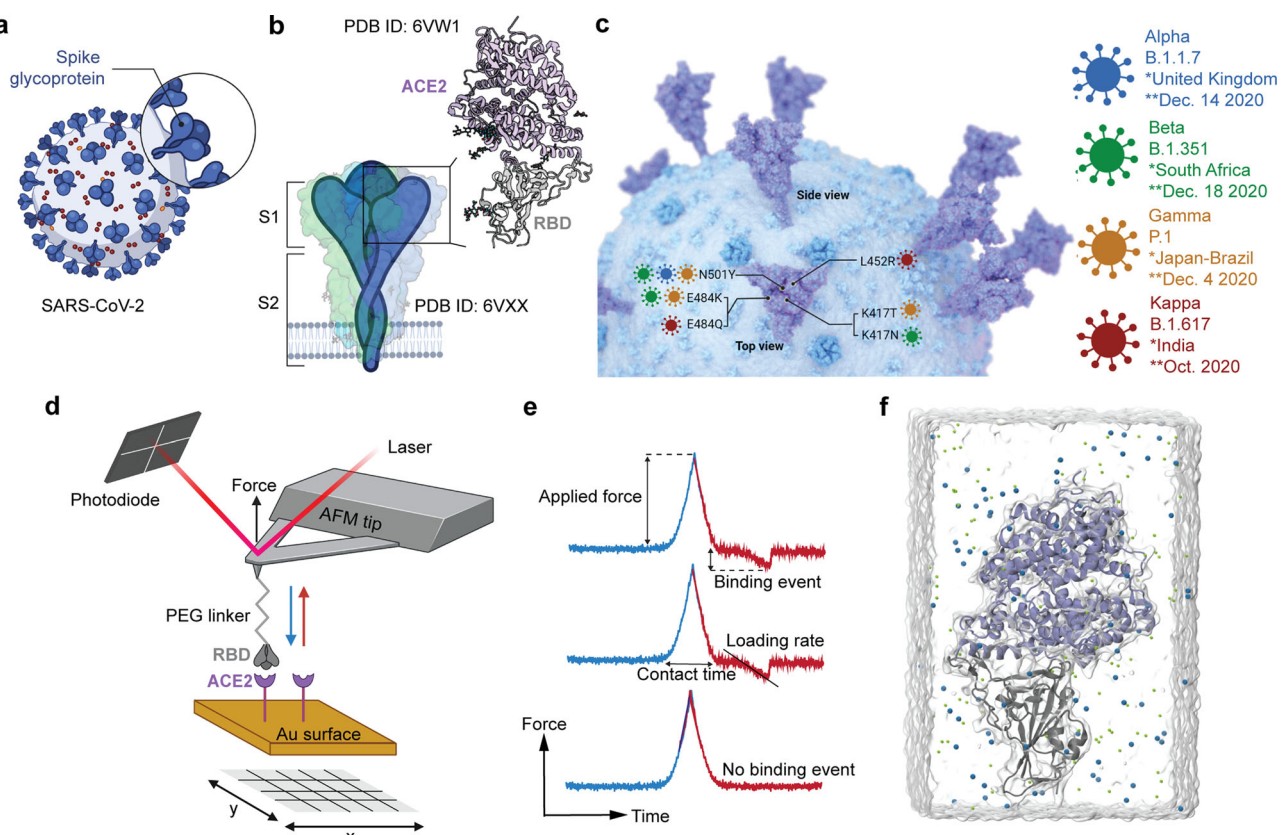

**Fig. 1 SARS-CoV-2 mutant binding to ACE host receptor quantified by atomic force microscopy and molecular dynamics simulation. a** Schematic of a SARS-CoV-2 virus particle, expressing at its surface the spike glycoprotein (S) that mediates the binding to host cells. **b** The S glycoprotein is composed of two subunits, S1 and S2, and is commonly represented as a sword-like spike. The Protein Data Bank (PDB) model of this glycoprotein (ID: 6VXX) reveals how the subunits are comprised of different regions that are fundamental to the infection process. Other Structural studies (PDB ID: 6VW1) have previously obtained a complex between the receptor-binding domain (RBD, a subunit of the S glycoprotein) and the angiotensin-converting enzyme 2 (ACE2) receptor. **c** Featuring a 3D rendering of SARS-CoV-2, this panel showcases the key spike protein mutations in the RBD domain on each of the studied SARS-CoV-2 variants of concern: Alpha, Beta, Gamma, and Kappa. **d** Schematic of probing RBD mutant binding to ACE2 receptors using atomic force microscopy (AFM). RBD is covalently attached to the AFM tip via a heterobifunctional PEG-linker and their binding to ACE2 receptors immobilized on a gold (Au) coated surface is probed. Pixel-for-pixel force distance (FD) curve-based AFM approaches and retracts the tip of an AFM cantilever from the sample to record interaction forces, $F$, over the tip-sample distance in FD curves. **e** Force−time curve from which the loading rate (LR) can be extracted from the slope of the curve just before bond rupture (LR = $\Delta F / \Delta t$) (upper curve). The contact time refers to the time when the tip and surface are in constant contact (middle curve). The lower curve shows no binding event. The tip approach is highlighted in blue, and tip retraction in red. **f** Representation of the system used in MD simulations.

whether this difference could originate from a higher binding affinity at the single-molecule level. Using FD-based AFM, we measured the strength of the RBD-ACE2 complexes by applying an external force ($F$) on the bond and extracted the kinetic properties of the interaction using the Bell−Evans model[18,19], which predicts that far from equilibrium, the binding strength of the ligand-receptor bond scales linearly with the logarithm of the loading rate (LR, force load on the bond over time). Experimentally, FD curves were recorded at various retraction speeds (Fig. 2b–f) and hold times (Fig. 2b–f, plots on the right) and plotted in so called dynamic force spectroscopy (DFS), showing the force vs LR. WT-RBD (Fig. 2b, black), Alpha-RBD (Fig. 2c, blue), Beta-RBD (Fig. 2d, green), Gamma-RBD (Fig. 2e, orange), and Kappa-RBD (Fig. 2f, purple) binding towards immobilized ACE2 receptors were analyzed, revealing that all complexes withstood forces between 25 and 400 pN over the range of applied LRs (N between 1000 and 2000 data points for each complex), in good agreement with the previously forces measured for the interactions probed between WT-S1—ACE2 and WT-RBD—ACE2[13] and for other virus-receptor bonds[17,20–23]. Although forces in this range can affect the conformational

stability of proteins, the RBD-ACE2 binding interface remained mechanically stable as evidenced by control experiments, showing thousands of stable interactions over several scans. Forces measured at different LRs were thoroughly analyzed and plotted in DFS plots (Fig. 2b–f), as established previously[13,17,22] (see "Methods" and Supplementary Figs. 1–5). Since single interactions were predominantly taking place, the data was fitted with the Bell−Evans model, and $k_{off}$ and $x_u$ were extracted for the probed interaction pairs. Similar distances to the transition state were observed for either the WT-RBD or the VoCs-RBD interacting with the ACE2 receptor, indicating a similar binding geometry/location of the binding pocket under these conditions. However, a decrease in the dissociation rate extrapolated to zero force was detected for the RBD mutants in the following order: WT-RBD ($k_{off} = 0.010 \pm 0.006$ s$^{-1}$) ≈ Alpha-RBD ≈ Beta > Kappa > Gamma, with the Kappa and Gamma forming twofold and fivefold more stable complexes, respectively. Accordingly, the lifetime $\tau$ of a single molecule bond can be directly obtained from the $k_{off}$ ($\tau = k_{off}^{-1}$), resulting in a bond lifetime of ≈100 ms for WT and Alpha-RBD—ACE2 complexes, ≈125 ms for Beta, ≈165 ms for Kappa, and ≈500 ms for Gamma. Higher forces

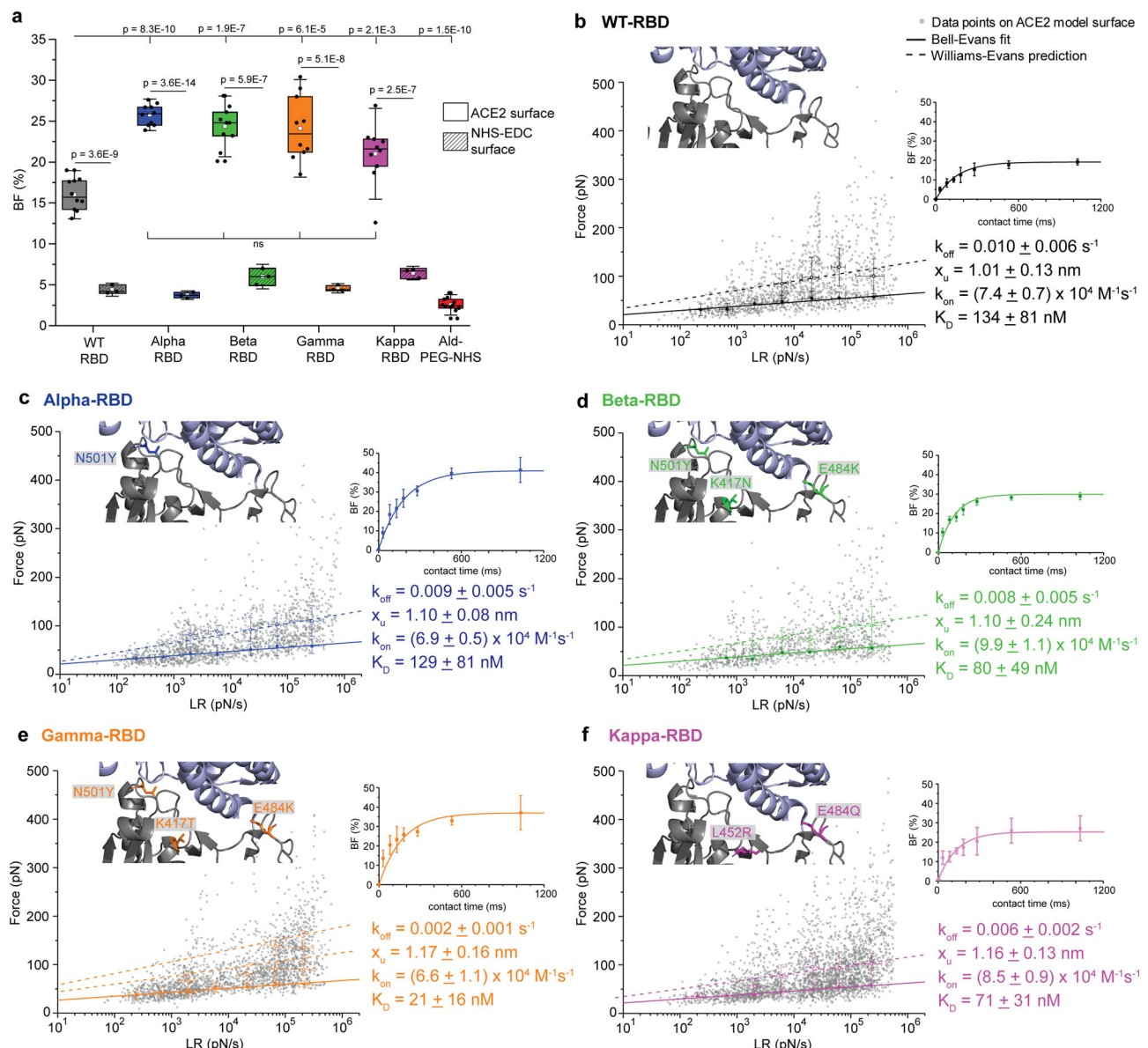

**Fig. 2 RBD mutant avidity for its ACE2 host receptor quantified by AFM. a** Box plot of specific binding frequencies (BF) measured by AFM between the functionalized tip (RBD mutants) and the grafted ACE2 model surface or the NHS-EDC surface, lacking ACE2 receptor (dashed boxes). One data point belongs to the BF from one map acquired at 1 μm s$^{-1}$ retraction speed and 250 ms contact time. The square in the box indicates the mean, the min/ max of the box the 25th and 75th percentiles respectively, and the whiskers the s.d. of the mean value. The line in the box indicates the median. $N = 10$ maps examined over 3 independent experiments for the ACE2 surface. $N = 3$ maps examined over 1–2 independent experiments for the NHS-EDC surface. $P$-values were determined by the two-sample $t$ test in Origin. **b–f** Dynamic force spectroscopy (DFS) plot showing the individual force extracted from individual FD curves (gray data points) as well as the average rupture forces, determined at seven distinct loading rate (LR) ranges measured either between ACE2 receptor and WT-RBD (**b**, $N = 1434$ data points), Alpha-RBD (**c**, $N = 1541$ data points), Beta-RBD (**d**, $N = 1094$ data points), Gamma-RBD (**e**, $N = 1952$ data points) or Kappa-RBD (**f**, $N = 2719$ data points). Data corresponding to single interactions were fitted with the Bell−Evans (BE) model (straight line), providing average $k_{off}$ and $x_u$ values. Dashed lines represent predicted binding forces for multiple simultaneous uncorrelated interactions ruptured in parallel (Williams−Evans [WE] prediction). Plots on the right: The binding frequency (BF) is plotted as a function of the contact time. Least-squares fits of the data to a mono-exponential decay curve (line) provide average kinetic on-rates ($k_{on}$) of the probed interaction. Further calculation ($k_{off}$/ $k_{on}$) leads to $K_D$. One data point belongs to the BF from one map acquired at 1 μm s$^{-1}$ retraction speed for the different contact times. All experiments were reproduced at least three times with independent tips and samples. The error bars indicate s.d. of the mean value. WT-RBD is colored in gray/black, and the VoCs Alpha-RBD in blue, Beta-RBD in green, Gamma-RBD in orange, and Kappa-RBD in purple.

observed on the DFS plots originate from the failure of uncorrelated bonds in parallel, as predicted by the William−Evans prediction. We observed a good correlation between the William −Evans prediction and the single-molecule data recorded. This observation underlines the important role of multivalency during virions attachment to cell surface receptors. The high density of

RBDs on virions (on the trimeric spikes and this one replicated many times on the virion surface), favors the establishment of multiple bonds towards an enhanced apparent lifetime of the virus−host interaction.

At the single-molecule level, a more accurate picture of the overall bond lifetime can be obtained by the dissociation constant $K_D$ of the

complex. For instance, high-affinity interactions have a long lifetime as $K_D$ is defined as the ratio between $k_{off}$ and the kinetic on-rate of the complex formation ($k_{on}$). Experimentally, by assuming that the receptor-bond complex formation can be approximated by a pseudo-first-order kinetics, $k_{on}$ can be extracted from the binding frequency (BF) measured at various hold times (Fig. 2b–f, plots on the right). This association rate depends on the effective concentration $c_{eff}$ described as the number of binding partners (RBD protein + ACE2 receptor) within an effective volume $V_{eff}$ accessible under free-equilibrium conditions. We approximated $V_{eff}$ by a half-sphere with a radius including the linker, RBD protein, and ACE2 receptor. For all the studied interaction pairs, we saw an exponential increase in the BF over time, and calculated their $k_{on}$, after extracting the interaction time. Collectively, these experiments lead to the following calculated equilibrium dissociation constants $K_D$ ($k_{off}/k_{on}$) in ascending order: Gamma ($K_D = 21 \pm 16$ nM) < Kappa ($K_D = 71 \pm 31$ nM) ≤ Beta ($K_D = 80 \pm 49$ nM) ≈ Alpha ($K_D = 129 \pm 81$ nM) ≈ WT ($K_D = 134 \pm 81$ nM). All these $K_D$ correspond to high-affinity interactions, confirming the high-affinity and stability of the complexes established by SARS-CoV-2 with its cognate ACE2 receptor. In particular, the Gamma variant and, to a lesser extent the Kappa variant, show lower $K_D$ values compared to the WT complex (see Supplementary Table 1). Other interaction studies between RBD WT and ACE2, as well as between SARS-CoV (80% sequence homology) and ACE2 reported specific, high-affinity association values in the nM range[13,24].

Taken together, our in vitro experiments show that the emerged VoCs, especially the Gamma and Kappa VoC, possess similar or better ACE2 receptors binding capacity compared with WT. $K_D$ values observed in this nM range imply a very strong affinity and specificity, making the development of anti-binding molecules targeting this interaction, such as antibodies, more difficult. The continuous appearance of VoCs seems to be an evolution towards complexes of higher stability while inserting key point mutations that could escape neutralization by convalescent sera or vaccine-derived sera.

**Energetic analysis and structural stability of RBD—ACE2 complexes.** To elucidate the molecular mechanism underlying the higher stability of RBD-ACE2 complexes observed for VoCs, we performed MD simulations. A cumulative 7.5 μs of all-atom MD simulation of fully solvated RBD—ACE2 complexes (Fig. 3a) were performed using the GROMACS package[25]. Standard modeling of WT and VoC systems, as well as MD simulation protocols, are reported in the Method section. Briefly, after the energy minimization step and equilibration of the simulation box, five replicas with 250 ns of unconstrained MD simulation for each system were performed and analyzed. The various RBD—ACE2 complexes were dynamic but remained stably bound throughout each simulation, as shown by Cα root mean square deviation (RMSD) from the starting equilibrated structure that was less than 10 Å for each RBD tested and the minor change of the root-mean-square fluctuation (RMSF) of VoCs with respect to WT (see Supplementary Fig. 6).

In the first step, we compared the total energy of the RBD—ACE2 complexes. For the WT complex, the potential energy between the RBD and ACE2, which is the summation of the intermolecular long-range Coulomb and Lenard—Jones interactions, showed an energy around −590.22 kJ/mol, in good agreement with previously published data[26,27]. Surprisingly, only the Kappa variant, among the four VoCs studied, showed a more stable binding complex with a decrease of the total energy of 15.2 kJ/mol (Fig. 3b). By looking at the interfacial energy, we observed on the contrary that only the three other VoCs (Alpha,

Beta, Gamma) show a gain in the Lennard—Jones energy of around 10 kJ/mol below the WT reference energy. This result is in good agreement with other recent computational studies[21,22] (Fig. 3c). This suggests that the RBD—ACE2 complexes for these three VoCs are stabilized by more intermolecular bonds such as hydrogen or vdW bonds. Several new intermolecular contacts were observed to form, break and reform during the simulations (Fig. 3d), mostly in the region around RBD residues 484 and 501, as can be observed in the enlarged regions (Fig. 3e, f).

Fine energetic analysis of the point mutations revealed important differences between the WT and VoCs (Fig. 4). Each individual mutation can influence binding towards ACE2 but can also be associated with local conformational and stability perturbations. Among all mutations, the mutation N501Y, which is common to three mutants (i.e. Alpha, Beta, and Gamma), seems to have the highest impact in terms of energies. This mutation strongly enhances the interaction between the RBD and ACE2 via new contacts which are correlated with a gain in the Lennard—Jones energy (Fig. 4j) of about 10 kJ/mol. The Kappa variant that lacks the N501Y mutation shows similar energy to the WT SARS-CoV-2. The mutation E484K in RBD changes the residue charge from a negative to a positive value and is present in two VoC (i.e. Beta and Gamma), while for the Kappa variant, it changes to uncharged glutamine. For this latter VoC, one weak interaction with the positively charged K31 residue in ACE2 appears in the MD simulation (Fig. 4g). The same contact is present in WT and Alpha and it vanishes in the case of Beta and Gamma, mostly due to the repulsive energy between both positive charged residues in RBD mutant and ACE2 (Fig. 3e, f). The positive charge residue K417 was mutated to an uncharged residue in Beta and Gamma (Fig. 4a, b). Figure 3d shows the contact K417-D30 that is found with a higher frequency (>0.7) for WT, Alpha and Kappa, and is compatible with a salt bridge. On the other hand, the contact with a mutation N417-D30 has a very low frequency (<0.3) for Beta and even it disappears for the Gamma VoC.

Furthermore, our analysis provides an additional picture associated with conserved residues in RBD whose chemical environment changes following mutations. The residue Y475 shows no change in total energy along WT and all mutants (Fig. 5c). However, the Lennard—Jones energy favors Alpha, Beta, and Gamma, over the Kappa mutant (Fig. 5d). This is manifested by the increasing contact frequency in Y475-S19 (~0.6) and a drop for Gamma and Kappa mutants around 0.4 (Fig. 5b). The energy of G496 residue shows the environment of Kappa as more favorable denoted by the lowest total energy, whereas its Lennard—Jones energy seems to fluctuate close to WT energy (Fig. 5e, f). For this residue two contacts are obtained (i.e. G496-D38 G496-K353) with high frequency (>0.6) (Fig. 5b) which are stabilized by long-range electrostatic interactions. Residue Q498 maintains a contact with K353 in ACE2 for WT and Kappa, while this contact is lost in the other mutants. This effect is correlated by having the local environment with lowest total and Lennard—Jones energies (Fig. 5g, h). Finally, Y505-E37 is a contact whose strength decreases in Alpha and Beta respect to WT and is recovered for Gamma and Kappa. Remarkably, we observed that for the Kappa variant, two contacts appeared at high frequency with the ACE2 β-hairpin (K353-G496 and K353-Q498) providing a larger and more stable interface between the ACE2 and RBD (Fig. 5i, j). We also observed that the majority of contacts are located close to the ACE2 β-hairpin within the RBD-ACE2 interface, which is even more stabilized in the Kappa variant (Fig. 5a). This process can be seen as key during spontaneous unbindings or under loading forces in AFM, as it might keep the ACE2 anchored on side of the interface for longer times.

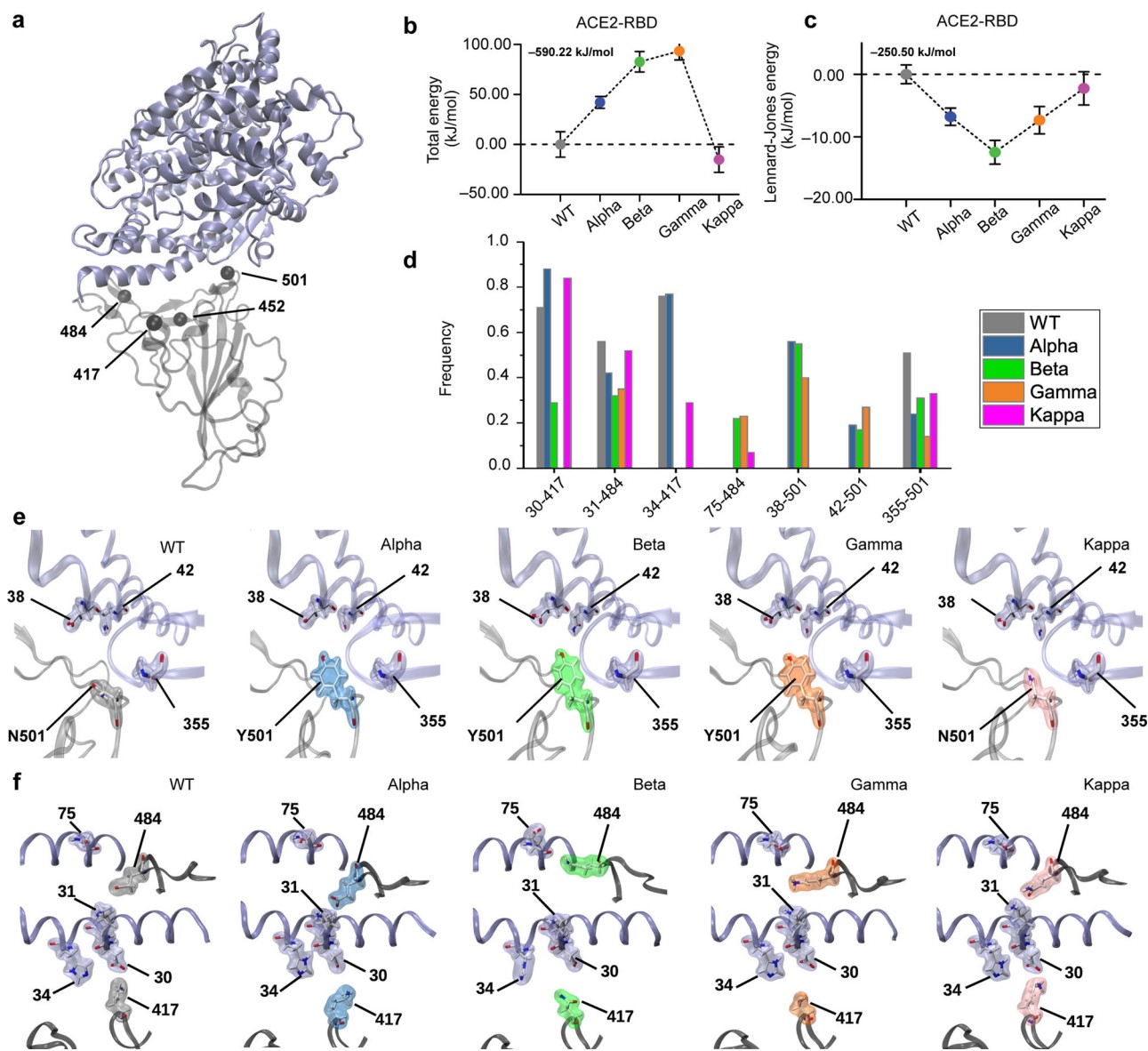

**Fig. 3 MD simulation of the RBD—ACE2 of WT and VoCs. a** Ribbon-like representation of the RBD—ACE2 complex and highlight of residues mutated in VoCs. **b** Total energies and **c** Lennard—Jones energies for all RBD—ACE2 complexes. **d** Average frequencies of contacts from 7.5 μs of MD trajectory of SARS-CoV-2 WT (gray) and VoCs Alpha (blue), Beta (green), Gamma (orange), and Kappa (purple). High-frequency contacts are shown in the form (ACE2 residue number)—(RBD residue number) for seven contacts involving relevant mutations. Zoom-in on the RBD—ACE2 region around **e** RBD residue 501 and **f** around RBD residues 484 and 417 for the WT and the 4 VoCs. Sidechains are represented by sticks. Residues 38, 42, and 355 in ACE2 contacting RBD residue 501 (**e**) and residues 30, 31, 34, and 75 in ACE2 contacting RBD 417 or 484, respectively, are shown. Data in **b**, **c** show the mean and the whiskers are the s.d. of the mean value obtained from $N = 3130$ (WT), $N = 2851$ (Alpha), and $N = 3130$ (Beta and Kappa).

**Mutated residues play a dynamic role for the RBD—ACE2 stabilization in VoCs**. Based on the single contact map information (Supplementary Tables 2–6) we can describe the relative importance for each mutated residue (i.e. 417, 452, 484, and 501) in VoCs. In Supplementary Table 7, we show the list of all contacts residue 417 forms with ACE2. In the WT, the salt bridge K417-D30 is part of the high frequency set with freq ∼ 0.7. The same contact is more persistent in Alpha variant with freq ∼ 0.9 and under mutations K417N and K417T in Beta and Gamma respectively, the contact frequency drops below 0.4, which correlates with the dramatic drop of the electrostatic contribution in the interfacial energy. For the Kappa variant, the salt bridge is re-established again with a higher freq ∼ 0.85.

Residue 452 does not establish any high-frequency (freq > 0.7) contact with the ACE2 receptor, however, it makes six high-

frequency intrachain RBD contacts (Supplementary Table 8) with residues 349, 350, 351, 492, and 493 in WT. In the Kappa variant, the fluctuation of those contacts is reduced making the RBD more stable. From the energetic characterization (see Fig. 4c, d) the flat profile of total and vdW energies for the WT RBD—ACE2 interface energy and VoCs is consistent with the absence of contacts mediated by this mutation in residue 452.

Residue 484 forms a very dynamic set of salt bridge interactions that explain the no dramatic increase of the total energy at the interface. In WT the E484 forms a stabilizing interaction with K31 denoted by a salt bridge and few strong hydrogen bonds (HBs) (Supplementary Table 2). Such interaction is not disrupted in the Alpha variant by mutation N501Y; instead, a stabilizing hydrophobic contact, Y501-Y41 within the ACE2 is reformed (Supplementary Table 3). For Beta, Gamma, and Kappa

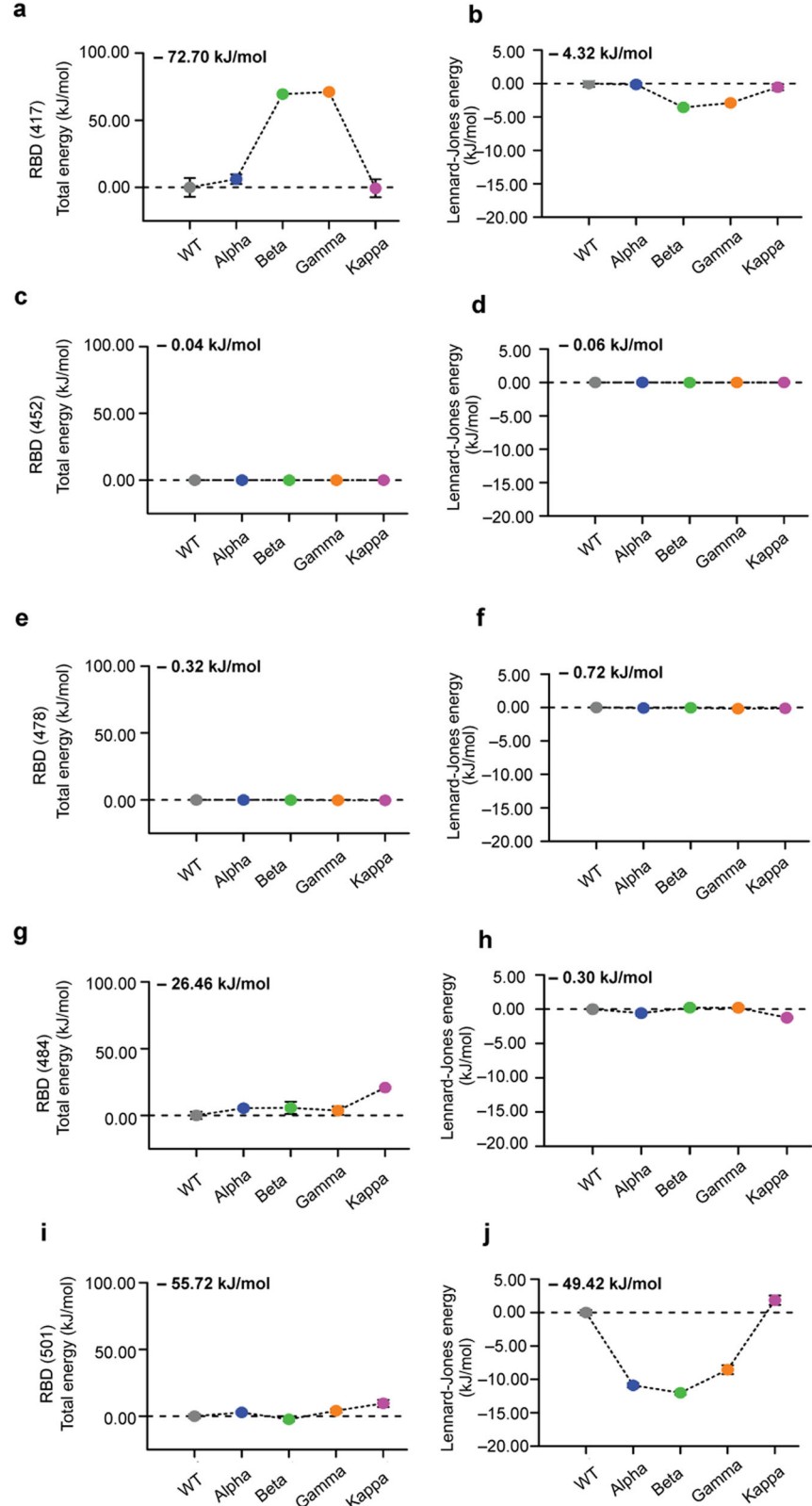

**Fig. 4 Energetic analysis of the RBD point mutations through MD simulation.** Total energies (**a**, **c**, **e**, **g**, **i**) and Lennard−Jones (**b**, **d**, **f**, **h**, **j**) interaction energies between contact ACE2 and RBD residues 417 (**a**, **b**), 452 (**c**, **d**), 478 (**e**, **f**) 484 (**g**, **h**) and 501 (**i**, **j**). For ease of comparison, the values have been reset to zero for the WT and the initial value of the WT is shown at the top left. Each point corresponds to the mean energy and the whiskers are the s.d. around the mean value obtained from $N = 3130$ (WT), $N = 2851$ (Alpha), and $N = 3130$ (Beta and Kappa). For most of the data point the error bars are very small and hidden in the symbol. WT is colored in gray, and the VoCs Alpha in blue, Beta in green, Gamma in orange, and Kappa in purple.

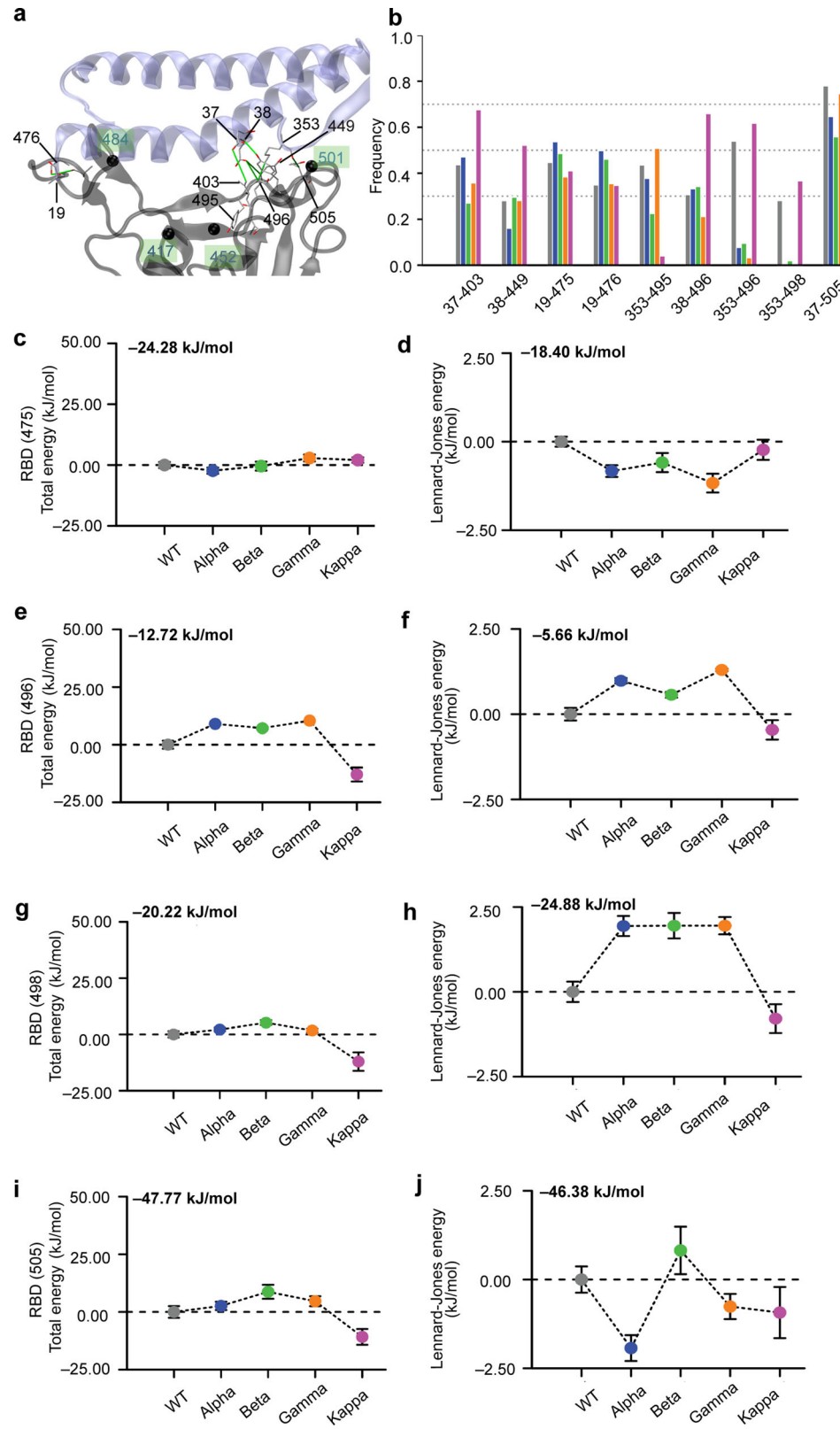

variants (Supplementary Tables 4–6), the salt bridge is lost under mutation of residue 484, however, a new salt bridge with ACE2 is formed, K484-E75 (see Supplementary Table 7) in Beta and Gamma variant, which offers an additional electrostatic stabilization at the RBD—ACE2 interface.

For the WT the residue 501 forms a high-frequency contact (freq ~ 0.8) with ACE2, denoted by N501-Y41. The same residue in Alpha, Beta, and Gamma variants forms a more stable hydrophobic contact due to the attractive π-π interaction with a higher frequency, 0.94, 0.97, 0.98 respectively, which is a sign of the interface stabilization as confirmed by vdW energy (see Fig. 4j). However, this contact becomes part of the low frequency (freq < 0.7) set in the Kappa variant (Supplementary Table 7). N501Y mutation is more dynamic and establishes more than one

**Fig. 5 Indirect structural and energetic changes of the conserved residues of the RBD—ACE2 complex due to VoCs point mutations. a** Ribbon-like representation of the RBD—ACE2 complex and highlights of the set of high-frequency contacts (solid green lines) that are formed by RBD conserved residues following mutations shown in black balls. **b** Average frequency of contacts between RBD and ACE2 molecules. These contacts are obtained from MD trajectories. Only contacts whose frequencies are larger than 0.3 and their difference to WT larger than 0.2 are shown. Bottom panels show total energies (**c**, **e**, **g**, **i**) and Lennard−Jones (**d**, **f**, **h**, **j**) interaction energies between contact ACE2 and RBD residues 475 (**c**, **d**), 496 (**e**, **f**), 498 (**g**, **h**) and 505 (**i**, **j**). Error bars in energies are given as standard deviations. Each point corresponds to the mean energy and the whiskers are the s.d. around the mean value obtained from $N = 3130$ (WT), $N = 2851$ (Alpha), and $N = 3130$ (Beta and Kappa). For most of the data point, the error bars are very small and hidden in the symbol. WT is colored in gray, and the VoCs Alpha in blue, Beta in green, Gamma in orange, and Kappa in purple.

extra contact in VoCs that are not present in the WT. As a summary, the mutation N501Y in the Alpha, Beta, and Gamma variants seems to strengthen the frequency of the Y501-Y41 contact.

**Antibodies targeting RBD of SARS-CoV-2 as potential treatment of COVID-19.** Since the S protein has been found to be the primary antigenic epitope on SARS-CoV-2, antibodies directed against this protein can neutralize the ability of the virus to bind and fuse with the target host cell[28,29]. The traditional neutralization mechanism is to block the receptor-binding site located between RBD and ACE2. Recently, several studies report the use of neutralizing mAbs against the RBD region of the S protein[30,31], rendering them a promising tool to be used as therapeutics against COVID-19. Here, we tested the binding inhibition efficiency of two IgG1 mAbs (B-K45 and B-R41, Diaclone SAS, France) directed against the WT RBD, using our single-molecule force spectroscopy approach (Fig. 6), with the goal to understand how they could neutralize the VoCs. As a reference, we first measured the BF between the WT-RBD or variants and the ACE2 receptor without mAb, using a contact time of 250 ms. Then, we injected the mAb at gradually increasing concentrations (1, 10, and 50 µg mL$^{-1}$) and pursued monitoring the BF (Fig. 6a). After B-K45 injection, we observed a progressive reduction of the BF as a function of the concentration for all WT and VoCs RBD (Fig. 6b–f). As a control for the specific inhibition capability of the mAB, we also tested a control mAb (B-D38, Isotype control IgG1, Diaclone SAS, France). Using this isotype control, we did not observe any specific inhibition confirming that the inhibition with B-K45 is specific. Quantitative comparison of inhibition level among RBDs (Fig. 6g) pointed out a BF reduction of >50% already at 1–10 µg mL$^{-1}$ for all the probed RBD–ACE2 interaction pairs, suggesting an IC$_{50}$ (50% inhibitory concentration) in the µg/mL range. Strikingly, this result shows that B-K45, although obtained from the immunization of mice with the WT-RBD, inhibits even more efficiently the VoCs-RBD binding to ACE2. A different behavior can be noticed for the second tested mAb, B-R41. Here, only binding of RBD WT and the Alpha variant to the ACE2 receptor can be efficiently inhibited (IC$_{50}$ at 1 µg mL$^{-1}$). The other RBD variants (Beta, Gamma, and Kappa) are still able to bind to the receptor in the presence of B-R41, however, with a 20–30% reduction of the BF. Altogether, our in vitro assays at the single-molecule level provide direct evidence that mAbs directed against WT-RBD can also inhibit the binding of the tested VoCs. However, as observed for the B-R41 mAb, VoCs can escape from this neutralization, probably as a consequence of the E484 mutation, as the three variants (beta, gamma, and kappa) showing a reduced inhibition share this particular mutation.

## Discussion

The emergence of SARS-CoV-2 variants of concern across the globe is worrisome, especially as the mutations could result in more transmissible viruses and impair recognition of the virus by

human antibody-mediated immunity. To better understand why these point mutations in the SARS-Cov-2 RBD led to the emergence of these variants, we studied how they affect the kinetic and thermodynamic properties toward ACE2 receptor binding. Using a combination of in vitro and in silico approaches, we linked the energetic properties of RBD-ACE2 complex formation to associated structural changes. By probing the interactions using AFM force spectroscopy (Fig. 2), we showed that mutations within the RBD of the different variants led in all cases to higher stability and affinity of the RBD—ACE2 complex. The data obtained here are consistent with a model in which variant spike proteins mediate increased transmissibility, at least in part, by enhancing ACE2 binding affinity[15,32].

Surprisingly, MD simulations revealed that only the Kappa mutant among all investigated variants show a more stable complex, as pointed out by the total energy extracted from the MD simulations (Fig. 3b). However, by looking at Van der Waals contribution of the interfacial energy, we noticed that the three other VoCs showed a more stable binding interface, with a gain of around 10 kJ/mol, stabilized by more intermolecular bonds. Looking at the residue level, a crucial role for the N501Y mutation present in the Alpha, Beta, and Gamma variants was identified, resulting in a significant gain in the stabilization of the RBD—ACE2 interface. Structurally, RBD residue 501Y, present in all three variants, has a large phenolic group that makes two additional contacts with ACE2 residues, coordinately stabilizing several segments within the RBD—ACE2 interface. Together with the AFM experiments, these results suggest that the destabilization of the interface enables the creation of additional key contact points around residue 501. We could see this as a 'key binding hotspot', providing a rationale for the decrease in $K_D$ observed by AFM. Regarding the Kappa variant, mutation of RBD residue 484 (E484Q) plays an important role from energetics point of view. Strikingly, the gain of energy results from long-range Coulomb interactions, establishing an electrostatic bridge at the RBD—ACE2 interface, and clamping the central zone of the complex, which limits the possibilities of conformational adjustments during the binding/unbinding process. This is in particular observed in Fig. 3d, which shows a reduction of the contact frequency for the Kappa VoC in the 'binding hotspot' area (around residues 501), compensated at least in part by additional contacts in the opposite part of the interface (around residues 417 and 484, Fig. 3f). MD simulations support the idea of a stabilized interface through multiple weaker molecular interactions located all over the interface. In this configuration, the gain in $K_D$ would not come from a "binding hotspot" but from a better-balanced interface from an energy point of view. Taken together, these observations demonstrate that these variants, although having arisen independently, evolve in such a way as to stabilize the contact interface between the RBD and the ACE2 receptor. This evolution appears to be towards hot binding spots making more stable contacts with the termini of the α-helix ACE2 receptor encompassing residues [22–57], confirming its great potential as therapeutic target[13]. We also noticed that the point mutations in variant RBDs result

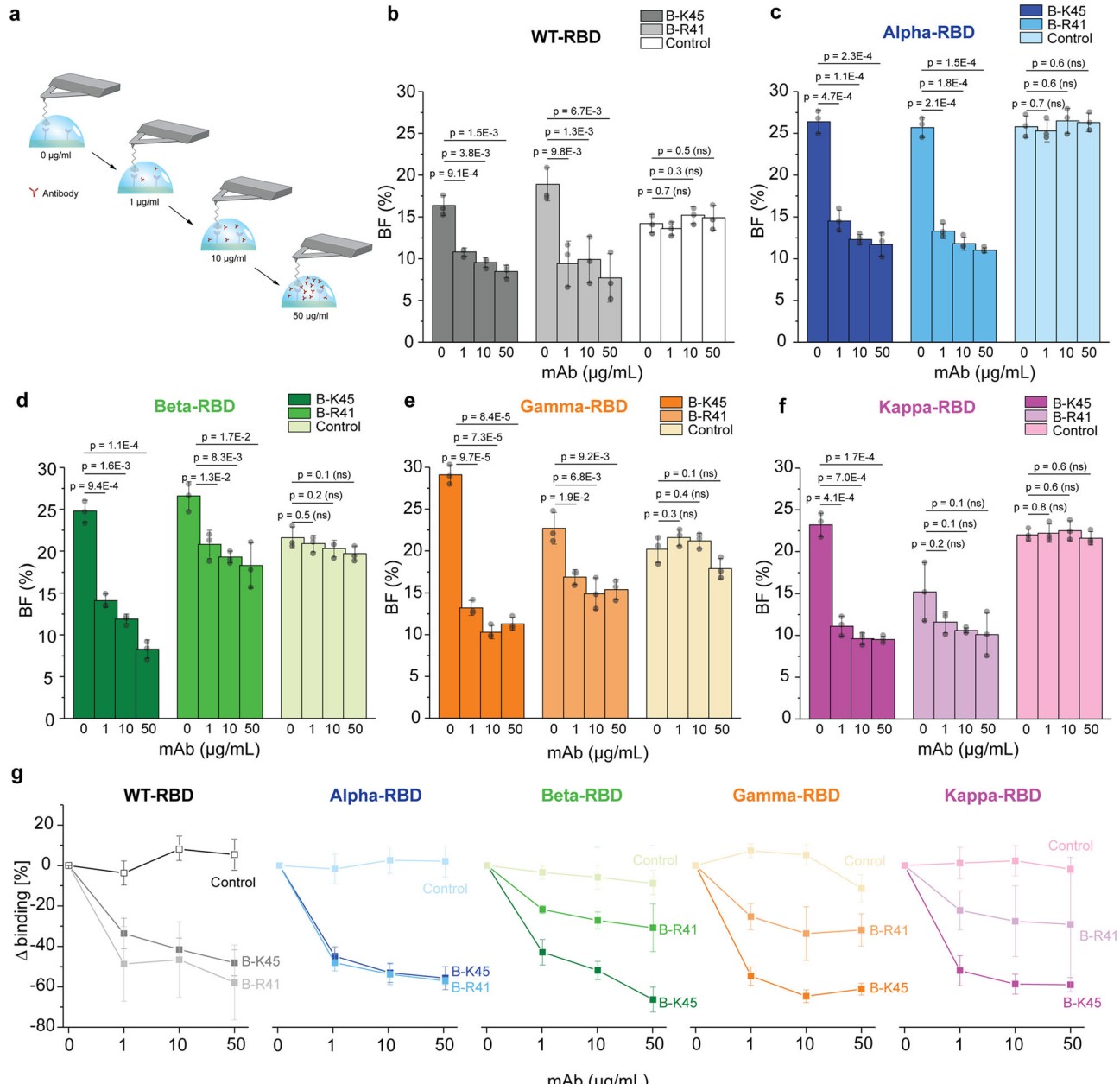

**Fig. 6 Probing antibody efficiency to inhibit binding of the different RBD mutants to ACE2 receptor. a** Efficiency of blocking antibodies is evaluated by measuring the binding frequency of the interaction between ACE2 and the RBD mutants before and after incubation with the three different monoclonal antibodies (mAbs) at increasing concentration (1–50 μg mL$^{-1}$). **b–f** Histograms, with individual data points overlaid [$N = 3$ independent experiments (tips and sample)], showing the binding frequency of the interaction between ACE2 and WT-RBD (**b**), Alpha-RBD (**c**), Beta-RBD (**d**), Gamma-RBD (**e**) or Kappa-RBD (**f**) without mAb and upon incubation with 1, 10, or 50 μg mL$^{-1}$ of two monoclonal anti-RBD (B-K45 and B-R41) or a control IgG1 isotype mAb. **g** Graph showing the reduction of the binding frequency. Data are representative of at least $N = 3$ independent experiments (tips and sample) per mAb concentration. *P*-values were determined by two-sample t-test in Origin. The error bars indicate s.d. of the mean value. WT-RBD is colored in gray/black, and the VoCs Alpha-RBD in blue, Beta-RBD in green, Gamma-RBD in orange, and Kappa-RBD in purple.

in structural rearrangement of conserved residues that stabilize more the binding interface. While AFM experiments provide only a global measurement, MD simulations allow a better understanding of the molecular mechanisms responsible for the stability of the interface between the RBD domain of SARS-CoV-2 and the ACE2 receptor. Out-of-equilibrium, and especially when these bonds are subject to external forces (AFM under our experimental conditions, shear forces in vivo, etc.), a better-balanced interface may be favorable to the bound state as it will require parallel rupture of multiple bonds, rather than serial breakage of the individual bond.

Finally, we also investigated the neutralization mechanism of mAbs at the single-molecule level. We compared two different mAbs and observed that one of those, the B-R41, lost its neutralization potential for the three SARS-CoV-2 variants exhibiting the most stable interface (Beta, Gamma, Kappa). On the opposite, the other mAb (B-K45) shows a similar inhibition efficacy for both, WT and VoCs. As those mAb have been obtained by the immunization of a mouse with the WT-RBD, our results demonstrate at the single-molecule level that new variants will have reduced susceptibility to neutralization by the polyclonal plasma antibodies of some individuals. However, it is tricky to

link the loss of neutralization susceptibility to a particular mutation. Nevertheless, we observed that the Alpha variant being neutralized by the B-41 shares the N501Y mutation with other mutant that escape the neutralization. This result is consistent with previous observations which suggest that N501Y mutation is unlikely to greatly affect neutralization by most human plasma, although it could contribute to increased viral titer or enhanced transmissibility[11,33].

Our mapping reveals broader features of the importance of mutations for RBD binding to the ACE2 receptor that are relevant to the evolution of SARS-CoV-2. It also suggests that there is an important area of future work to understand how viral mutations can impact the stability of the interface and use this knowledge to develop strategies to robustly counteract viral antigenic evolution.

## Methods

**Protein expression and purification**. All antibodies used in the study (1, 10, and 50 µg mL$^{-1}$ in PBS buffer) were developed and purified by Diaclone SAS (France). Anti-RBD mAbs (B-K45 and B-R41) come from the immunization of BALB/c mice with WT-RBD protein (Diaclone SAS, 715-H16-0BU). B-D38 (Diaclone, SAS, France) has been used as an isotype control IgG1.

**Functionalization of AFM tips**. MSCT-D cantilevers (Bruker) were used to probe the interaction between S1 spike glycoprotein subunits (Diaclone SAS, 715-H16-0BU, 715-H24-0BU, 715-H22-0BU, 715-H21-0BU, 715-H20-0BU) and the extra-cellular domain of ACE-2 receptors (Diaclone SAS, 715-H19-0BU). AFM tips were functionalized with the spike proteins using NHS-PEG$_{24}$-Ph-aldehyde (Broad-pharm) linkers as described previously[34]. The cantilevers were rinsed by immersing in chloroform for 10 min and further cleaned by UV-radiation and ozone. Sila-nization was performed in the gas phase[35] by placing the AFM tips in a desiccator with separate vials containing 3-aminopropylriethoxysilane APTES (30 µL) and triethylamine TEA (10 µL). The tips were incubated for 2 h, following which the desiccator was flushed with argon and left to cure for at least two days. Following amino-functionalization, AFM tips were coupled with flexible PEG linkers. The cantilevers were immersed in a solution of NHS-PEG$_{24}$-Ph-aldehyde (3.3 mg in 0.5 mL of chloroform) in the presence of triethylamine (30 µL). After 2 h of incubation at room temperature, the cantilevers were thoroughly washed with chloroform three times and dried under nitrogen.

For AFM tips functionalized with RBD proteins, 50 µL of protein solution (0.1 mg mL$^{-1}$ in PBS pH 7.4) was added to the cantilevers placed Parafilm inside a Petridish. To this, sodium cyanoborohydride NaCNBH$_3$ (1 µL of 1 M stock solution) was added and was kept for 1 h at room temperature. Then, ethanolamine hydrochloride (2.5 µL of 1 M stock solution, pH 8.0) was added to block the free aldehyde groups on the AFM cantilever. Subsequently, the AFM tips were washed three times with PBS buffer and used immediately for the experiments. The whole tip functionalization protocol was carried out at RT.

**Preparation of ACE2 coated model surfaces**. ACE2 protein (Diaclone, SAS, 715-H19-0BU) was immobilized on gold-coated surfaces using NHS-EDC chemistry. Gold-coated surfaces were rinsed with ethanol, dried under nitrogen, cleaned for 15 min by UV-ozone treatment, and incubated overnight in a solution containing alkanethiol (99% 11-mercapto-1-undecanol 1 mM (Sigma Aldrich) and 1% 16-mercaptohexadecanoic acid 1 mM (Sigma Aldrich) in ethanol). Resultant surfaces were washed with ethanol, dried under a gentle stream of nitrogen, and immersed in a solution 25 mg mL$^{-1}$ of dimethylaminopropyl carbodiimide (EDC) and 10 mg mL$^{-1}$ of N-hydroxy succinimide (NHS). Finally, the chemically activated samples were incubated with ACE2 protein (25 µL, 0.1 mg mL$^{-1}$ in PBS, pH 7.4) on parafilm for 30 min and washed with PBS buffer and used on the same day. The whole surface preparation protocol was carried out at RT.

**FD-based AFM on model surfaces**. FD-based AFM on model surfaces was performed at room temperature in PBS using functionalized MSCT probes (Bruker, nominal spring constant of 0.030 N m$^{-1}$ and actual spring constants measured prior to the experiment using thermal tune method[36]). A Bruker Nano8 operated in force-volume (contact) mode in fluid (Nanoscope software v9.1) was used for the determination of kinetic on-rate (measuring the binding probability for different contact times of 0, 50, 100, 150, 250, 500, and 1000 ms). For each force map, a scan size of 5 µm, a set point force of 500 pN, resolution of 32 × 32 pixels, and a line frequency of 1 Hz were used.

A JPK Force Robot 300 AFM was used to measure the loading rates and disruption forces by DFS analysis (using a constant approach speed of 1 µm s$^{-1}$ and variable retraction speeds of 0.1, 0.2, 1, 5, 10, and 20 µm s$^{-1}$) (Control Software v6.1.144). All experiments were carried out at RT.

DFS data were extracted using the JPK Data Analysis software (JPK, v7.1.14+) and further analyzed using Origin software (OriginLab, version 2019) to fit

histograms of rupture force distributions for distinct LR ranges, and to apply various force spectroscopy models[13,16,17]. Only binding events between at rupture distances between 12 and 30 nm were considered as specific events and selected for extracting unbinding force and loading rate[13]. For kinetic on-rate analysis, the binding probability (fraction of curves showing binding events) was determined at a certain contact time ($t$) (the time the tip is in contact with the surface). Those data were fitted and $K_D$ calculated as described previously[37]. In brief, the relationship between interaction time ($\tau$) and BP is described by the following equation:

$$\mathrm{BP} = A \times \left[1 - \exp\left(\frac{-(t - t_0)}{t}\right)\right] \quad (1)$$

where $A$ is the maximum BP and $t_0$ the lag time. Origin software is used to fit the data and extract $\tau$. In the next step, $k_{on}$ was calculated by the following equation, with $r_{eff}$ the radius of the sphere, $n_b$ the number of binding partners, and $N_A$ the Avogadro constant

$$k_{on} = \frac{\frac{1}{2} \cdot 4\pi r_{eff}^3 \cdot N_a}{3\eta_b \tau} \quad (2)$$

The effective volume $V_{eff}$ ($4\pi r_{eff}^3$) represents the volume in which the interaction can take place.

**Antibody inhibition assays**. To study the influence of antibodies on the binding affinity between RBD of different mutants and the ACE2 receptor, binding probabilities were measured before and after the addition of three different antibodies (1, 10, and 50 µg mL$^{-1}$ in PBS buffer) of three different antibodies and measured thereafter at RT. Briefly, three force-volume maps were recorded on three different areas as described previously in the absence of any antibody (i.e., force−volume mode, 1 µm s$^{-1}$ approach and retraction speed, map size of 5 µm, threshold force of 500 pN, 32 × 32 pixels, 512 samples per line, frequency of 1 Hz, and contact time of 250 ms on the surface). Thereafter, antibodies were added to the fluid cell and three maps were recorded for each concentration.

**Modeling of SARS-CoV-2 RBD—ACE2 complex and RBD mutants**. The (ACE2) is recognized by the receptor-binding domain (RBD) of SARS-CoV2 spike (S) protein and together they form a protein complex (PDB code 6M0J for WT)[38]. The RBD and ACE2 comprise residues 333−526 and 19−615 respectively. The models in this work retain the Zn$^{2+}$ ions which are electrostatically bonded to residues HIS 374, HIS 378, GLU 402 in ACE2. The protocol to model mutations in the RBD is made by replacing amino acids in an energetic favorable conformation by UCSF Chimera[39] and Dunbrack rotamer library. The Alpha variant includes only one mutation (i.e. N501Y) in their RBD domain, while Beta has three mutations (i.e. K417N, E484K, and N501Y). Also, the Gamma variant has also three mutations (i.e. K417T, E484K, and N501Y) while the Kappa variant presents only two mutations (i.e. L452R and E484Q) in their RBD domain.

**All-atom MD simulation**. The molecular package GROMACS[25] was used to carry out all-atom MD simulations. The protein, water, and ions were modeled by CHARMM36m[40] force-field. The solvent employed the TIP3P water model. The WT and VoCs RBD in complex with ACE2 were optimized using standard energy minimization (1500 steps) and conjugate gradient (500 steps) algorithms available in GROMACS. Also, the equilibration of systems was done in the NVT and NPT ensemble. The v-rescaled thermostat was used to maintain the temperature constant in the NVT ensemble for 10 ps and the Berendsen barostat in the NPT ensemble for 100 ps. Later the production was carried out in the NVT ensemble. The RBD—ACE2 complex was initially placed at the center of the simulation box which was solvated in 12 Å octahedral water shell. MD trajectories were checked for atomic clashes. All simulations used 0.15 mol L$^{-1}$ of NaCl molecules. We employed a 2 fs timestep for every MD simulation. During pre-production restraints of 41.84 kJ mol$^{-1}$ Å$^{-2}$ were applied on the backbone of the complex. The last 50 ns of unconstrained MD simulation in the NVT ensemble using the Parrinello−Rahman barostat maintained the system at 300 K and 1 bar. The production runs include five replicas for the WT and 4 variants. Each replica was run by 250 ns. Thus, we got a cumulative time of 1.5 µs for each system. Trajectories were saved every 40 ps/frame, which were used in further analysis (see below).

**Energetics and contact analysis of the RBD—ACE2 complex for WT and mutants**. The energetic analysis of RBD—ACE2 interface employs the gmx energy utility in GROMACS. We calculate the non-bonded contribution given by electrostatic and Van der Waals energies originated by the Coulomb interaction and the Lennard−Jones potential respectively for each replica. The total number of frames used for the analysis equals 54005 in each 1.5 µs MD trajectory. These energies consider amino acids residue in RBD and ACE2 that constitute the interface. Here we called total energy the non-bonded energy contribution at the RBD—ACE2 interface, given by $E$(interface) = $E$(RBD + ACE)-$E$(RBD)-$E$(ACE2), where $E$(RBD + ACE) is the non-bonded energy of the whole complex and $E$(RBD) and $E$(ACE2) is the single energy contribution for each single protein. In addition, we employ our contact analysis methodology already validated in several protein complexes[41,42] and recently validated for the study of the S protein[43,44]. The

contact maps are based on the OV + rCSU approach[45]. We obtained from our trajectories 19377 contact maps which have been employed in the analysis of the frequency per contact.

**Reporting summary**. Further information on research design is available in the Nature Research Reporting Summary linked to this article.

## Data availability

The Source data underlying Figs. 2–6 are provided as a Source Data file. The source data for the protein structures in Fig. 1 are available at the RCSB protein database under the following references 6VW1 and 6VXX. All other relevant data are available from the corresponding authors upon reasonable request. Source data are provided with this paper.

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

## Acknowledgements

This work was supported by the Université catholique de Louvain, the Foundation Louvain, and the Fonds National de la Recherche Scientifique (FRS-FNRS). This project received funding from the European Research Council under the European Union's Horizon 2020 research and innovation program (grant agreement No. 758224) and from the FNRS-Welbio (Grant # CR-2019S-01). The funders had no role in study design, data collection, and analysis, decision to publish, or preparation of the paper. M.K. and D.A. are postdoctoral researcher and research associate at the FNRS, respectively. A.P. acknowledges financial support from the National Science Center, Poland, under grant 2017/26/D/NZ1/00466, the grant MAB PLUS/2019/11 from the Foundation for Polish Science, and also computational resources were supported by the PL-GRID infrastructure. Cartoons in Figs. 1a–d and 6a were created with BioRender.com.

## Author contributions

M.K., A.R and D.A. conceived the project, planned the experiments, and analyzed the data. M.K., A.R. and B.J. conducted the AFM experiments. R.A.M. and A.B.P. designed, performed, and analyzed the MD simulation. All authors wrote the paper.

## Competing interests

The authors declare no competing interests.
