## [Peer Review File · Nature Communications]

Molecular insights into receptor binding energetics and neutralization of SARS-CoV-2 variantsREVIEWER COMMENTS

Reviewer #1 (Remarks to the Author):

The manuscript describes a very interesting and currently highly relevant quantification of the influence of currently worldwide occurring mutations in the Receptor Binding Domain (RBD) or Receptor Binding Motif (RBM) of the spike protein of SARS-CoV-2 on the binding behavior to the native receptor ACE2 of the host cell. Experimental studies are performed on a model system using the isolated S1 subunits of the mutants and the isolated ACE2. Quantification of binding affinity is performed by AFM. The data are analyzed by MD simulations and based on that by a detailed molecular/atomic breakdown of the energetic basis of the interaction between RBM and ACE2. Furthermore, the influence of two different antibodies on the binding behavior is investigated experimentally by AFM. The research group of David Alsteens has a proven and very well documented expertise regarding AFM studies. In principle, the study can be a valuable addition to the understanding of the influence of mutations on the binding behavior of SARS-CoV-2. However, not least due to the partly incomprehensible conclusions of the authors, several fundamental questions remain unanswered, which could also be answered within the framework of this study.

Major concerns

1. Based on the koff and KD values in Fig. 2b-f, I cannot understand the statements in the abstract

"We observe a direct link between increased RBD—ACE2 complex stability and the greater transmissibility observed for the variants of concern."

and on page 10, line 185,

"Especially the Gamma variant as well as the Kappa and Beta ones show significantly lower KD values compared to the WT complex."

Based on the experimental values, this statement is only true for the Gamma mutant, possibly still for the Kappa mutant, but not for the other mutants, especially considering the s.d. values. See also the following points.

2. In the same line, page 10, line 190:

"Taken together, our in vitro experiments show that the four investigated VoCs, having emerged, show higher binding properties to ACE2 receptors, both in terms of affinity and stability."

By reasons given above this statement is not comprehensible.

3. Based on Fig. 2a, all mutants have a higher binding frequency than the wild type. However, this cannot be deduced from inserts Figs. 2b (wild type) and 2f (Kappa mutant) because here the binding frequencies and their dependencies are very similar.

4. The break-off histograms are interpreted to mean that mono-, di-, trivalent, etc. bonds have been formed. This implies that there must be multiple binding sites on the AFM tip. It is not clear whether and to what extent this was taken into account when calculating the on rate from the BF curves, i.e., what value for n_b was actually used. Technically, surely the physically available number of interaction pairs would have to be taken into account, but not those that then actually form bonds in the time available (and thus become visible in the force measurement).

In this context, I miss the relevant information on the surface density of ACE2 on the AU-surface, on its reproducibility in independent experiments and on the comparison of the surface density on the natural host cell surface. The same applies to the density of S1 on the AFM-tip and the comparison to the intact virus.

5. It is completely unclear how the experimental data were fit to Bell-Evans. As I understand it, the spectrograms are supposed to be composed of monovalent and higher orders. It is not obvious how these individual orders were separated from each other. For the most part, the histograms show

rather broad distributions and only very rarely structures that look like a series of individual peaks. These separations are critical, since the Bell analysis strongly depends on them. How this was done with the relatively structureless distributions presented here and how it can be concluded (page 9, line 164)

"We observe a good correlation between the William-Evans prediction and the single-molecule data recorded."

is not explained. The occurrence of two dashed lines in Fig. 2e is not explained.

6. The MD simulations cannot satisfactorily explain the experimental results, because according to them either the Kappa mutant (Fig. 3b) or the Beta mutant (Fig. 3c) would bind most strongly to ACE2. However, according to the experimental results, this would be the Gamma variant. Based on the calculation of total energy (Fig. 3b), one would expect poorer binding of the other mutants compared to the wild type except for the Kappa variant.

However, the authors refer to the interfacial energy in their conclusions (Fig. 3c). An explanation of the difference between total energy and interfacial energy is necessary for readers who do not have the necessary knowledge in this field. The question arises whether the values of interfacial energy, which are rather in the range of RT , are sufficient to explain stronger binding at all. As explained below, there are significant differences between the experimental and theoretical studies which, in my view, are essential to discuss and address the question of the extent to which the results are comparable.

7. Based on the fact that the interfacial energy is in the range of RT , the question of the role of multivalence also arises for the stable binding of the virus, i.e. even weakly increased binding energies can lead to a more stable binding of the virus via multivalence. This idea should be considered in the Discussion.

8. The experimental results and those obtained by MD are analyzed and discussed only from the point of view of the direct interaction of residues between RBM and the ACE2-binding site. To what extent this allows the direct comparison of the experimental and theoretical results is questionable. In the case of the MD simulations, only the interaction between RBM and ACE2-binding site was considered, but not the entire S1 subunit. In the case of the MD simulation, an up-conformation is always assumed; in the case of S1 subunit, it cannot be assumed that only the up-conformation is present. In the discussion, the fact of an equilibrium and kinetics between up- and down-conformation is not considered at all. Even if this should not play a role for the S1 subunit, a discussion of the relevance of the results presented here for the native trimer would then be necessary in any case. In the trimer, up- and down-conformations play an important role with respect to binding to ACE2.

9. It is known that mutations in the spike protein have an influence on the balance between up- and down-conformation. What influence do the mutations presented here have on the balance between up- and down-conformation and what consequences does this have for the interpretation of the experimental results?

10. Related to point 8, the use of different terms for the S1 subunits used in the experiments is confusing, in particular it blurs the differences between experiment and MD.

The following terms are used: S1, spike protein, RBD, RBM.

Even in Fig. 1, it is not clear that S1 was used for the experiments and not just RBD (Fig. 2d), especially since Fig. 2b shows the entire trimer.

11. An interesting aspect of the study is the reduction in binding after pre-incubation of S1 with two different antibodies. Surprising is the finding that already after the first concentration level a further increase of the antibody concentration up to fivefold had only a marginal effect on the binding and did not lead to an almost complete inhibition. Assuming multivalent binding of the viruses to the host cell, the question arises whether virus binding and thus infection would be affected at all. The

interpretation of the results also seems biased to me. Could it be that the antibodies not by direct binding to the RBM, but by binding in the vicinity of the RBM cause a reduction in binding by e.g. steric hindrance, but not complete inhibition? Also, the question arises whether the balance between up- and down-conformation is affected. The authors should discuss these aspects in the discussion.

As far as I could see from Diaclone's catalog, ACE2 is present as a monomer in the experiments described here. Naturally, ACE2 exists as a homodimer. I miss a reference to this in the present study. Are there known differences in binding to the RBM between ACE2 monomer and dimer? If so, how do these affect the interpretation of the results?

12. Data on measurement temperature and incubation temperatures are missing.

13. Similarly, there is no indication of how long the samples were incubated with the antibodies at the respective concentration levels and what the duration of the incubation times are based on.

14. Line 122 mentions that only events occurring at a peak-to-sample distance > 12 nm are counted. The selection of curves requires a more detailed description, e.g. whether a maximum distance was considered.

15. Line 146 states that the force spectra for each complex are composed of $N > 2500$ data points. This is, according to the SI, the case only for Kappa. For the others, it is 1000 - 2000 data points.

16. Line 149 states that the proteins could be unstable given the applied forces, however this would have been refuted by controls in this study. Selected control experiments could still be shown in SI and at the same time explain/quantify the statement "stable interactions over several scans". However, my understanding is that the control experiments only show that the binding and thus the RBD structure are stable, however, no statement can be made about the stability of the whole S1 structure.

17. p. 11, line 204
Use 'FD' instead 'F-D'

18. p. 27 line 455:
Correct (xx

This manuscript by Koehler et al represents a new application of AFM for understanding the RBD-ACE2 binding of SARS-CoV-2 mutants. This methodology has been already used with this virus by the same group and represents a nice example of how single molecule techniques can give responses to biomedical problems. However, I have some concerns that should be considered before publication.

1. Figure 2a shows that mutants exhibit higher binding frequencies than WT. However, mutants do not show dramatic differences between them. If each mutant corresponds to a different structure of the spike, what is the power of this force spectroscopy to resolve the effects of different mutants, beyond the clear difference with WT? I would include the control experiments of fig. S1 in figure 2a to gain clarity in this important figure.
2. Charts of k_{on} and k_{off} , extracted from figure 2:

These charts do to seem to depict strong differences between WT and mutants. Can the authors explain the importance of the fig. 2b-f fittings to understand the differences between spikes? Mutants seem very similar to WT.

3. The contact time for obtaining data of fig. 2a is 250 ms. Now, if we go to the same contact time of fig. 2b-f insets, there is a lack of correspondence between data. In particular, in fig. 2a WT shows most of BF values above 15%. but the inset of fig. 2b shows many more below 15%. Something similar happens in beta, gamma and kappa mutants: the values of fig. 2a do not agree with the insets. While alpha shows similar BFs, Kappa exhibits the worst correspondence: ~22% in fig. 2a and ~12% in the inset of fig. 2f.
4. The manuscript also shows very meritorious theoretical simulations that disentangle the energetics of ACE2-RBD mutants binding. However, I do not see any direct connection between theory and experiments beyond the qualitative explanation of saying that mutants are worse than WT. Specifically, I would expect a feedback between theory and experiments in the sense of obtaining direct experimental/theoretical parameters that could be used reciprocally. Theory and experiments seem two different histories that might be published in different papers.

Reviewer #3 (Remarks to the Author):

In this manuscript the authors used both experimental atomic force microscopy (AFM) and MD simulation to investigate the effect of known SARS-COV-2 variants (alpha, beta, gamma and kappa) on the kinetics, thermodynamics and structural properties of RBD-ACE2 complex. They also tested the neutralization efficiency of two mAbs against these variants and found that one mAb shows excellent anti-binding properties against all variant while the other lost its neutralization for three variants having E484 mutation. They performed an accumulative 7.5 μ s MD simulation on RBD-ACE2 complex and its variants and this will be the focus of this review. Two mutations (N501Y and E484Q) are found to be important for higher stability of the complex. The authors computed the Leonard-Jones (LJ) and total energy of the complex interface to provide a description of local and long-range effects caused by mutations. Moreover, the authors provided information about residue contacts during MD simulation to describe the stabilizing or destabilizing effects of mutations on pair of residues. I have the following comments:

1. Can the authors present specific fluctuations in residues computed as root mean square fluctuations (RMSF) and how it differs between variants in different parts of the interface between RBD and ACE2. This could be average RMSF between replicas.
2. The Kappa variant (L452R, E484Q) showed a more stable complex with a decrease in total energy (15.2KJ/mol). The authors discussed the role of E484Q mutation in stabilizing the complex but the role of L452 mutation is not discussed in detail. Does the mutation to a charged residue change the contact of L452 residue. How does it affect the vdw interaction of L452 with the corresponding residue in ACE2.
3. Residue E484 in RBD is close to residue K31 in ACE2. Therefore, one would expect the mutation E484K to have a negative effect on the binding energy in the complex. However, this change is not dramatic as shown in figure 4g. Could the authors explain the reason behind this as this mutant is mostly important for being an antibody escape mutation.
4. K417 mutation is thought to have a high impact on the binding, since it disrupts the salt-bridge with D30 on ACE2. Is this the main reason for higher energy of Beta and Gamma complexes relative to other mutations? What is the relative effect of mutation in K417 with respect to for example residue N501. A comparison between mutation in different residues would help to understand their relative importance.
5. How do these mutations affect the structure of the complex in terms of H-bonds and salt-bridges? A more structural picture of the complex is missing in the manuscript. What hydrophobic contacts are present in the WT and mutant complexes and how does the mutations affect different type of interactions. For instance, N501 in WT is in contact with residue Y41 on ACE2. Mutation N501Y can thus have a pi-pi stacking.
6. Total energy and total LJ energy of the system are not descriptive in terms of the effect of mutations on the binding free energy. The authors should not use this simple metric to represent interaction as it is not related to the thermodynamic quantity driving stability, i.e. free energy. It is standard and accepted that free energy calculation methods such as end-point MMPBSA/MMGBSA should be used to determine the binding free energies in protein-protein complexes. The authors should use more complex methods for binding free energy computation.
7. Fig4 the y-label for both left and right columns are the same.

Point-by-Point Response to the Reviewers Comments

Reviewer #1 (Remarks to the Author):

The manuscript describes a very interesting and currently highly relevant quantification of the influence of currently worldwide occurring mutations in the Receptor Binding Domain (RBD) or Receptor Binding Motif (RBM) of the spike protein of SARS-CoV-2 on the binding behavior to the native receptor ACE2 of the host cell. Experimental studies are performed on a model system using the isolated S1 subunits of the mutants and the isolated ACE2. Quantification of binding affinity is performed by AFM. The data are analyzed by MD simulations and based on that by a detailed molecular/atomic breakdown of the energetic basis of the interaction between RBM and ACE2. Furthermore, the influence of two different antibodies on the binding behavior is investigated experimentally by AFM. The research group of David Alsteens has a proven and very well documented expertise regarding AFM studies. In principle, the study can be a valuable addition to the understanding of the influence of mutations on the binding behavior of SARS-CoV-2. However, not least due to the partly incomprehensible conclusions of the authors, several fundamental questions remain unanswered, which could also be answered within the framework of this study.

Authors: Thank you for your encouraging and constructive comments. Below we have explained point-by-point how we have addressed your questions and concerns to strengthen our manuscript for resubmission in *Nature Communications*.

1) Based on the koff and KD values in Fig. 2b-f, I cannot understand the statements in the abstract "We observe a direct link between increased RBD—ACE2 complex stability and the greater transmissibility observed for the variants of concern." and on page 10, line 185, "Especially the Gamma variant as well as the Kappa and Beta ones show significantly lower KD values compared to the WT complex."

Based on the experimental values, this statement is only true for the Gamma mutant, possibly still for the Kappa mutant, but not for the other mutants, especially considering the s.d. values. See also the following points.

Authors: We agree with the reviewer and clarified both sentences. Several studies demonstrated an increased transmissibility for the VoCs^{1,2}. In a recent publication³, direct correlation between the mutation in the VoCs and the higher transmissibility has been described. Following the reviewer's comment, we looked again at our data and we observe that the K_D of the Gamma, Kappa and to lesser extent Beta variants are decreasing. The Alpha variant does not show any difference in terms of affinity, although a higher transmissibility is

observed. This shows that a direct link between affinity and transmissibility is not correct and we have adapted the text accordingly. Additionally, since reviewer 2 raised similar concerns, we performed some statistical analysis to compare the k_{on} , k_{off} and resulting K_D values of the VoCs with the ones from the wildtype (see **Fig. R3, reviewer 2, comment 2**). As it can be seen, the Gamma and Kappa mutant differ significantly in their K_D value compared to the WT, pointing out their increased affinity towards ACE2 binding.

The sentence in the abstract (lines 30-31) now reads: "We observed for several variants of concern a significant increase in the RBD—ACE2 complex stability."

On line 196, the text has been modified to: "In particular, the Gamma variant and, to a lesser extent, the Kappa and Beta variants, show significantly lower K_D values compared to the WT complex."

2) In the same line, page 10, line 190: "Taken together, our in vitro experiments show that the four investigated VoCs, having emerged, show higher binding properties to ACE2 receptors, both in terms of affinity and stability." By reasons given above this statement is not comprehensible.

Authors: We agree with the reviewers and modified the sentence (now line 201) to: "Taken together, our in vitro experiments show that the VoCs that emerged generally possess better ACE2 receptors binding capacity compared with WT, both in terms of affinity and stability."

3) Based on Fig. 2a, all mutants have a higher binding frequency than the wild type. However, this cannot be deduced from inserts Figs. 2b (wild type) and 2f (Kappa mutant) because here the binding frequencies and their dependencies are very similar.

Authors: The authors thank the reviewers for pointing out this mismatch in **Fig 2**. We have added a revised version of **Fig. 2** in page 11 of the main text.

We reevaluated the binding probability for the Kappa VoC, with three different functionalized AFM probes and collected 3072 data points for each contact time. Each data point represents a discrete single force measurement at any position on the sample surface. We report that there was a slight increase (~25%, observed in total) in the binding frequency in comparison to our last reported value (~20%). Despite the slight gain in binding frequency, the k_{on} and K_D values obtained [$k_{on} = (8.5 \pm 0.9) \times 10^4 \text{ M}^{-1}\text{s}^{-1}$; $K_D = 71 \pm 31 \text{ nM}$] remained similar to our previous reported values [$k_{on} = (8.4 \pm 0.2) \times 10^4 \text{ M}^{-1}\text{s}^{-1}$; $K_D = 71 \pm 25 \text{ nM}$]. Additionally, the combined dataset (**Fig. R1**) yields very similar results [$k_{on} = (8.7 \pm 0.8) \times 10^4 \text{ M}^{-1}\text{s}^{-1}$; $K_D = 69 \pm 29 \text{ nM}$], showing high level of data reproducibility. The binding frequency depends greatly on the accessibility of the binding partners on the tip that may vary from one tip to another. However, the kinetic parameters are not significantly affected by the position of the ligand onto the tip. Here, our additional experiments provide evidence that even with different binding frequencies observed, the extracted k_{on} remains very similar.

Figure R1 | Comparison of the data sets for the BP vs. contact time experiments on the interaction between Kappa-RBD and ACE2.

4) The break-off histograms are interpreted to mean that mono-, di-, trivalent, etc. bonds have been formed. This implies that there must be multiple binding sites on the AFM tip. It is not clear whether and to what extent this was taken into account when calculating the on rate from the BF curves, i.e., what value for n_{b} was actually used. Technically, surely the physically available number of interaction pairs would have to be taken into account, but not those that then actually form bonds in the time available (and thus become visible in the force measurement). In this context, I miss the relevant information on the surface density of ACE2 on the AU-surface, on its reproducibility in independent experiments and on the comparison of the surface density on the natural host cell surface. The same applies to the density of S1 on the AFM-tip and the comparison to the intact virus

Authors: We thank the reviewers for giving us an opportunity to explain our data treatment procedure. To gain a full understanding of the energy landscape of the single-molecule interaction, several force distributions are probed as a function of the loading rate (LR, force applied over time). This gives a scatter plot known as the DFS plot represented in **Fig. 2b-f** in the main text. To determine the valency of the interaction between ACE2 and RBD of different VoCs, every single data point was analyzed through distinct ranges of LR, which was plotted as a force histogram and eventually fitted with multi-peak Gaussian distribution as established previously. Using this distribution, the most probable rupture force was calculated. The

individual force histograms as a function of LR ranges are shown in **Supplementary Figs. 1-5**, showcasing that mostly single-bond interaction was taking place. In the higher loading rate-ranges, there is evidence of double-peak or triple-peak Gaussian suggesting the formation of two and three bonds in parallel. This can be explained on the basis that ACE2 is dimeric, implying that two S1 or RBD molecules can interact at the same time.

The surface density of ACE2 on gold substrates has been optimized to ensure sparsely distributed individual receptors on the surface and to maximize the probability to record a single rupture event. According to the theoretical framework of Tees et al.⁴, binding probabilities of less than 10% are considered to be in the perfect range for probing single molecule interactions, which is the fundamental aim of this study. Since the radius of the AFM tip curvature is not so low that only one molecule can be attached to the apex, it is likely that 2 (or 3) molecules might be attached which can lead to multiple bond ruptures. Following this proposed framework, the expected probability of probing single-molecule interactions in comparison to multiple interactions follows a Poisson distribution, where the fraction of events that involve rupture of N bonds equals to:

$$P(N) = \frac{\left(\frac{1}{P_{tot}} - 1\right) \left[\ln\left(\frac{1}{1 - P_{tot}}\right)\right]^N}{N!}$$

where P_{tot} is the overall probability of detecting rupture events. Consequently, fractional probability of rupturing more than one bond P_{mult} is

$$P_{mult} = 1 - \left(\frac{1}{P_{tot}} - 1\right) \ln\left(\frac{1}{1 - P_{tot}}\right)$$

This equation indicates that multiple bond ruptures are expected to occur rather often in DFS measurements if P_{tot} is high. For example, if $P_{tot} = 0.5$, then the multiple bond ruptures occur only approximately twice less as often as the single bond ruptures. According to the second equation, for multiple bond ruptures to constitute less than 5% and for all bond ruptures the overall probability P_{tot} should be less than 10%, which matches perfectly the 5-15% binding probability measured for the ACE2 –RBD interaction at 0 s contact time. Practically speaking, this means that our tip- and surface immobilization strategies as well as experimental parameters are perfectly tuned in order to probe single molecule interactions.

5) It is completely unclear how the experimental data were fit to Bell-Evans. As I understand it, the spectrograms are supposed to be composed of monovalent and higher orders. It is not obvious how these individual orders were separated from each other. For the most part, the histograms show rather broad distributions and only very rarely structures that look like a series of individual peaks. These separations are critical, since the Bell analysis strongly depends on them. How this was done with the relatively structureless distributions presented here and how it can be concluded (page 9, line 164) "We observe a good correlation between the William-Evans prediction and the single-molecule data recorded." is not explained. The occurrence of two dashed lines in Fig. 2e is not explained.

Authors: With respect to the theory of force spectroscopy, by applying the Bell-Evans model (only valid for single-bond rupture) we can extract the kinetic parameters of single bonds (straight line in Fig. 2b-f, actual fit of the data) and use the predictive Williams-Evans model to determine whether the multiple bonds are correlated or not, i.e., if all the bonds are equivalent or if the first binding event induces an allosteric modulation of the second binding event. Therefore, if the maxima of the second, third, ... peak of the Gaussian distribution (**Fig. R2**) overlay with the Williams-Evans prediction (dashed line in **Fig. 2b-f**), then we can conclude that multivalent interactions are established as uncorrelated bonds loaded in parallel. This is showcased in **Supplementary Figs. 1-5**.

Figure R2 | Representative force histogram of one distinct loading rate range.

6) The MD simulations cannot satisfactorily explain the experimental results, because according to them either the Kappa mutant (Fig. 3b) or the Beta mutant (Fig. 3c) would bind most strongly to ACE2. However, according to the experimental results, this would be the Gamma variant. Based on the calculation of total energy (Fig. 3b), one would expect poorer binding of the other mutants compared to the wild type except for the Kappa variant. However, the authors refer to the interfacial energy in their conclusions (Fig. 3c). An explanation of the difference between total energy and interfacial energy is necessary for readers who do not have the necessary knowledge in this field. The question arises whether the values of interfacial energy, which are rather in the range of RT , are sufficient to explain stronger binding at all. As explained below, there are significant differences between the experimental and theoretical studies which, in my view, are essential to discuss and address the question of the extent to which the results are comparable.

Authors: Thanks for the opportunity to complement our molecular description from the point of view of the all-atom MD simulation. We agree with the reviewer about the comparison between simulations and experiments. We did not aim for a one-to-one comparison as the time and length scales are very different, actually our computational study provides a comprehensive description of the underlying interactions responsible for the stabilization of

the RBD/ACE2 VoCs. In principle, **Fig 3b and 3c** reported the contribution of the RBD/ACE2 interface for the total energy and for the VdW contribution respectively. Each figure is discussed in the MS and supports the relevance of the electrostatic interaction for the interface stabilization (e.g. salt-bridges and hydrogen bonds). **Fig. 3c** shows the VdW contribution given in kJ/mol which is 10-fold smaller than the electrostatic contribution. For instance, the relative energy of a native interaction in proteins is about 6.276 KJ/mol⁵. This means by looking at the VdW energy, we may infer the gain of 1 or 2 additional native contacts in a given variant with respect to the WT. **Fig. 3b**, shows a substantial difference in “local” interface energy. For the Alpha variant, the mutation N501Y affects uncharged residues G496 and Q498 by making new contacts with charged ACE2 residues D38 and K353. This creates an electrostatic repulsion as shown in **Fig. 3c**, while in the WT we detected 2 salt-bridges (i.e. D30-K417 and K31-E484) at the interface. For Beta and Gamma, the electrostatic energy decreases mostly due to mutation E484K as it introduces a positive charge whose effect is repulsive, in case of Kappa the same effect is played by L452R which adds one positive charge, but here the other mutation E484Q weakens the overall repulsion effect by reestablishing one of the WT salt bridge (i.e. D30-K417) and other hydrogen bonds (HB) in the RBD segment (500-505), offering a new stabilization. A direct comparison with experiment is not trivial and for that purpose we may need to employ computational demanding free energy calculation to calculate the binding free energy (ΔG_0) that is equal to, $\Delta G_0 = G_{\text{bind}} - G_{\text{unbind}}$. Based on thermodynamics, a simple relationship between ΔG_0 and the binding contact (K_b) holds,

$$\Delta G_0 = -k_b T \ln(K_b), \text{ with } K_b = 1/K_D.$$

Here, K_D is the dissociation constant and according to our SMFS data, the values range in decreasing order as follows: WT>Alpha>Beta>Kappa>Gamma and inversely in binding free energy. Such methodologies work well in protein-ligand characterization but they are very sensitive for large protein complexes.

We have improved the discussion of the calculation of the interface energy from MD simulation (see in method section, lines 545-548). The value of VdW interface energy is about 15 KJ/mol (**Fig. 3b**) while room temperature (RT) is about 2.34 KJ/mol. Certainly, following the argument provided above, we noticed the formation of additional contacts such as 2 HBs in the region 500-505. As reported in **Fig. 5b** and according to our additional analysis of HB contacts (**Supplementary Tables 1-5**) and the characteristic energy per contact in protein in MD simulation (i.e. 6.276 KJ/mol). The presence of additional contacts supports the stabilization of the RBD/ACE2 complex in VoCs.

7) Based on the fact that the interfacial energy is in the range of RT, the question of the role of multivalence also arises for the stable binding of the virus, i.e. even weakly increased binding energies can lead to a more stable binding of the virus via multivalence. This idea should be considered in the Discussion.

Authors: As we mentioned, the aim of the energies computed at the interface of the RBD/ACE2 complex and denoted as the interfacial energy is to characterize the chemical

environment and correlate it with structural changes such as the appearance/disappearance of contacts at the interface. Thus, our energy estimator and the binding free energy are not comparable in energy scale for the protein complex. However, the interfacial energy and contact analysis add information which can be correlated with stability of the variants in SMFS experiments. Several other studies^{6,7} have discussed the binding free energy picture of the RBD/ACE2 complex of VoCs with respect to WT. As a result, the value of ΔG varies in the range of -60 to -100 kcal/mol for WT and VoCs. In practice, the multivalence issue is not possible for the RBD/ACE2 complex at RT, as the corresponding binding free energy is 100-fold larger than RT. To avoid confusion about the energy scales presented in our manuscript, we have improved the discussion of the energy scales and binding energies in the revised manuscript (line 399).

8) The experimental results and those obtained by MD are analyzed and discussed only from the point of view of the direct interaction of residues between RBM and the ACE2-binding site.

To what extent this allows the direct comparison of the experimental and theoretical results is questionable. In the case of the MD simulations, only the interaction between RBM and ACE2-binding site was considered, but not the entire S1 subunit. In the case of the MD simulation, an up-conformation is always assumed; in the case of S1 subunit, it cannot be assumed that only the up-conformation is present. In the discussion, the fact of an equilibrium and kinetics between up- and down-conformation is not considered at all. Even if this should not play a role for the S1 subunit, a discussion of the relevance of the results presented here for the native trimer would then be necessary in any case. In the trimer, up- and down-conformations play an important role with respect to binding to ACE2.

Authors: For the computational studies we carried out a quantitative description of the molecular changes at the single level of interactions for the RBD/ACE2 interface of WT and VoCs. As the reviewer pointed out the spike protein is constituted by S1 and S2 domains and as it undergoes conformational changes from closed to open state which is key for viral recognition of the cell receptor. As we mentioned above, our additional study also describes the chemical character of the most relevant interactions formed at the RBD/ACE2 interface. Such information is calculated in equilibrium conditions and it helps us to understand the dynamic role played by mutations in terms of appearance/disappearance of contacts and the correlation with the kinetic studies of the SMFS.

Additionally, we apologize for a mistake in Fig. 1b, leading to confusion. In the previous version, we mistook the position of S1 and S2 in the S glycoprotein. A revised **Fig. 1** is now added to the manuscript (page 6).

9) It is known that mutations in the spike protein have an influence on the balance between up- and down-conformation. What influence do the mutations presented here have on the balance between up- and down-conformation and what consequences does this have for the interpretation of the experimental results?

Authors: Thanks for the opportunity to clarify further the effect of mutations in the RBD VoCs. The reviewer is correct that several other variations in the distal region of the spike protein and near the S1/S2 such as the well-known D614G⁸⁻¹⁰ can facilitate transitions toward the open state and thus enhancing recognition with the ACE2 receptor. The role played by the mutations positioned at the receptor-binding module (RBM) which is part of the RBD is to strengthen the interaction with ACE2. Thus, they do not engage in conformational changes of the spike protein. We have clarified this message in the revised manuscript.

10) Related to point 8, the use of different terms for the S1 subunits used in the experiments is confusing, in particular it blurs the differences between experiment and MD. The following terms are used: S1, spike protein, RBD, RBM. Even in Fig. 1, it is not clear that S1 was used for the experiments and not just RBD (Fig. 2d), especially since Fig. 2b shows the entire trimer.

Authors: The authors thank the reviewer for pointing out this mistake and the necessary changes have been made in Fig. 1.

11) An interesting aspect of the study is the reduction in binding after pre-incubation of S1 with two different antibodies. Surprising is the finding that already after the first concentration level a further increase of the antibody concentration up to fivefold had only a marginal effect on the binding and did not lead to an almost complete inhibition. Assuming multivalent binding of the viruses to the host cell, the question arises whether virus binding and thus infection would be affected at all. The interpretation of the results also seems biased to me. Could it be that the antibodies not by direct binding to the RBM, but by binding in the vicinity of the RBM cause a reduction in binding by e.g. steric hindrance, but not complete inhibition? Also, the question arises whether the balance between up- and down-conformation is affected. The authors should discuss these aspects in the discussion.

Authors: Recognition between antigen and antibody is driven by an interplay of many forces like the non-covalent interaction forces, Van der Waals forces, salt-bridge interaction etc. In a recently reported article¹¹, it was found that the major contribution to neutralization is driven by induction of steric hindrance due to binding of the antibody to the ACE2, which provides steric competition and direct competition for interface residues which in turn reduce the binding probability.

As far as I could see from Diaclone's catalog, ACE2 is present as a monomer in the experiments described here. Naturally, ACE2 exists as a homodimer. I miss a reference to this in the present study. Are there known differences in binding to the RBM between ACE2 monomer and dimer? If so, how do these affect the interpretation of the results?

Authors: We thank the reviewer for raising this important point. The reviewer is correct that the ACE2 receptor exists in a homodimeric state and the interaction forces between the RBD and ACE2 will vary between its monomeric and a dimeric state. We would like to emphasize that in our study we focus on the single-molecular monomeric state of ACE2 and all the interaction forces arising from a single pair of ACE2-RBD. This has been clarified on line 123.

12) Data on measurement temperature and incubation temperatures are missing.

Authors: The authors thank the reviewer for pointing out this mistake and the necessary changes have been made in the methods section (lines 474, 485, 503-505).

13) Similarly, there is no indication of how long the samples were incubated with the antibodies at the respective concentration levels and what the duration of the incubation times are based on.

Authors: The authors thank the reviewer for the remark. In the manuscript, all experiments were conducted under physiological conditions (PBS buffer) and at room temperature. The details have been added to the main text in lines 503-505.

The blocking experiments performed in the presence of antibodies were done without any incubation time. After measuring the initial binding frequency in the absence of blocking antibodies, the buffer in the AFM fluid cell was exchanged with a solution containing antibodies, sequentially starting from 1 $\mu\text{g mL}^{-1}$ until 50 $\mu\text{g mL}^{-1}$. Changes have been made in lines 503-505.

14) Line 122 mentions that only events occurring at a peak-to-sample distance > 12 nm are counted. The selection of curves requires a more detailed description, e.g. whether a maximum distance was considered.

Authors: This distance criterion separating nonspecific from specific interactions ensures that the specific adhesion event originates from the ligand at the free end of the stretched PEG-polypeptide linker. The 12 nm takes into account the stretching of the PEG spacer as well as the size of the folded RBD protein.

15) Line 146 states that the force spectra for each complex are composed of $N > 2500$ data points. This is, according to the SI, the case only for Kappa. For the others, it is 1000 - 2000 data points.

Authors: We apologize for this mistake and the corrected the value accordingly (line 156).

16) Line 149 states that the proteins could be unstable given the applied forces, however this would have been refuted by controls in this study. Selected control experiments could still be shown in SI and at the same time explain/quantify the statement "stable interactions over several scans". However, my understanding is that the control experiments only show that the binding and thus the RBD structure are stable, however, no statement can be made about the stability of the whole S1 structure.

Authors: We thank the reviewer for this comment and clarified this point in line 160, where we specified that the RBD-ACE2 binding interface remained mechanically stable over several scans. Based on previous reports¹², we compared the binding of only the RBD domain and the S1 and did not observe any change in the binding. Of course, we cannot ensure that local unfolding of S1 occurs during the force-probing, but as the recorded interactions are similar

for the RBD domain only and the whole S1 and in good agreement with other techniques (SPR /BLI¹²), we are certain that the applied force does not affect the RBD-ACE2 binding interface.

17) p. 11, line 204: Use 'FD' instead 'F-D

Authors: The authors thank the reviewer for pointing out this mistake and the necessary changes have been made (line 217).

18) p. 27 line 455: Correct (xx

Authors: The authors thank the reviewer for pointing out this mistake and the necessary changes have been made.

Reviewer #2 (Remarks to the Author):

This manuscript by Koehler et al represents a new application of AFM for understanding the RBD-ACE2 binding of SARS-CoV-2 mutants. This methodology has been already used with this virus by the same group and represents a nice example of how single molecule techniques can give responses to biomedical problems. However, I have some concerns that should be considered before publication.

Authors: Thank you for your encouraging and constructive reviews. Below we have explained point-by-point how we have addressed your specific comments and concerns.

1) Figure 2a shows that mutants exhibit higher binding frequencies than WT. However, mutants do not show dramatic differences between them. If each mutant corresponds to a different structure of the spike, what is the power of this force spectroscopy to resolve the effects of different mutants, beyond the clear difference with WT? I would include the control experiments of fig. S1 in figure 2a to gain clarity in this important figure.

Authors: The authors thank the reviewer for this comment. Majority of single-point mutations have little to no-effect on the global secondary structure of the proteins, but the site-specific perturbations can induce a change in a binding affinity between ligand-receptor pairs by the modulation of the local framework of non-covalent interactions. We have already reported this in our recent paper where we studied the interaction between human G protein-coupled PAR1 receptor using FD-AFM and found that single-point mutations have an important bearing on the ligand-binding process¹³. AFM in this regard, is a highly sensitive nanoscopic tool that can resolve forces with piconewton (pN) sensitivity. Following the referee's suggestion, we included the control experiments of previous Supplementary Figure 1 in the new, revised **Fig. 2a** to provide clarity in this important figure.

2) Charts of k_{on} and k_{off} , extracted from figure 2:

These charts do seem to depict strong differences between WT and mutants. Can the authors explain the importance of the fig. 2b-f fittings to understand the differences between spikes? Mutants seem very similar to WT.

Authors: We thank the reviewer for pointing out this critical point and apologize for the confusion. We performed statistical tests to compare the k_{on} , k_{off} and resulting K_D values of the VoCs with the ones from the wildtype (see **Fig. R3**). As it can be seen, the Gamma and Kappa mutants differ significantly in their K_D value compared to the WT, pointing out their increased affinity towards ACE2 binding. This conclusion has been also questioned by reviewer 1, comment 1 and changes in the main manuscript have been made accordingly (lines 30, 168-171, 192-194, 196).

Figure R3 | Statistical analysis of k_{on} , k_{off} and resulting K_D values of the VoCs compared to WT values.

3) The contact time for obtaining data of fig. 2a is 250 ms. Now, if we go to the same contact time of fig. 2b-f insets, there is a lack of correspondence between data. In particular, in fig. 2a WT shows most of BF values above 15%. but the inset of fig. 2b shows many more below 15%. Something similar happens in beta, gamma and kappa mutants: the values of fig. 2a do not

agree with the insets. While alpha shows similar BFs, Kappa exhibits the worst correspondence: ~22% in fig. 2a and ~12% in the inset of fig. 2f.

Authors: Thanks for pointing out this mismatch in the data representation. As it was already explained in comment 3, reviewer 1, we performed the binding probability experiment again for the Kappa VoC, with three different functionalized AFM probes. We observed a slightly higher value of binding frequency (~25%) which is quite in the range as observed for the other VoCs. We have rectified **Fig. 2a** and then added the new BF vs. contact time plot in **Fig 2f**. This slight discrepancy in data could be attributed to a combination of lesser density of RBD on the AFM tip and AFM scan area.

4) The manuscript also shows very meritorious theoretical simulations that disentangle the energetics of ACE2-RBD mutants binding. However, I do not see any direct connection between theory and experiments beyond the qualitative explanation of saying that mutants are worse than WT. Specifically, I would expect a feedback between theory and experiments in the sense of obtaining direct experimental/theoretical parameters that could be used reciprocally. Theory and experiments seem two different histories that might be published in different papers.

Authors: We appreciate the opportunity to further discuss our theoretical results in the revised MS. To avoid confusion, we have described the determination of the non-bonded energy associated with the RBD/ACE2 interface in our studies (see Method section). Based on Fig. 3b we report the change in total non-bonded energy in VoCs with respect to WT. Here, it is clear Alpha, Beta and Gamma variants seem to be affected by a severe change in the electrostatic energy rather than a change in VdW energy. In order to gain an insight on the issue of the gained stability owned by the VoCs with respect to WT. We have extended our contact map analysis at the single interaction level for each system and report a quantitative analysis of the chemical character of the stabilizing interactions such as polar, hydrophobic, salt bridge interactions and the strength of hydrogen bonds established at the RBD/ACE2 interface (see **Supplementary Tables 1-6**). This new information allows a better comparison between WT and VoCs. As a result, we show the gain of polar interactions in Kappa with respect to WT and the strengthening of pre-existing contacts in the RBD segment (500-505) in VoCs. Furthermore, the formation of two new contacts K353-T496 and K353-C498 reported in **Fig. 5b** support the idea of the stabilization in the Kappa variant.

Reviewer #3 (Remarks to the Author):

In this manuscript the authors used both experimental atomic force microscopy (AFM) and MD simulation to investigate the effect of known SARS-COV-2 variants (alpha, beta, gamma and kappa) on the kinetics, thermodynamics and structural properties of RBD-ACE2 complex. They also tested the neutralization efficiency of two mAbs against these variants and found that one mAb shows excellent anti-binding properties against all variant while the other lost its neutralization for three variants having E484 mutation. They performed an accumulative 7.5 μ s MD simulation on RBD-ACE2 complex and its variants and this will be the focus of this review. Two mutations (N501Y and E484Q) are found to be important for higher stability of the complex. The authors computed the Leonard-Jones (LJ) and total energy of the complex interface to provide a description of local and long-range effects caused by mutations. Moreover, the authors provided information about residue contacts during MD simulation to describe the stabilizing or destabilizing effects of mutations on pair of residues. I have the following comments.

Authors: We thank the reviewer for this positive and encouraging report on our work. Below we explain point-by-point how we have addressed the specific points in our revised manuscript.

1) Can the authors present specific fluctuations in residues computed as root mean square fluctuations (RMSF) and how it differs between variants in different parts of the interface between RBD and ACE2. This could be average RMSF between replicas.

Authors: We appreciate the reviewer's suggestion and we have calculated the RMSF profiles for the WT and VoCs (**Supplementary Figure 6**). As one can see the effect of the mutation N501Y in the Alpha variant is to increase the stiffness in several parts of the RBD by reducing residue fluctuations without disturbing the overall binding. For Beta and Gamma variants, the change in stiffness is negligible compared to Alpha variant. For the kappa variant, the mutation E484Q increases the flexibility around residue 484 compared with WT whereas L452R does not play a major role in the whole RMSF. As an observation from MD simulation we noticed that mutations do not severely affect the ACE2 binding region which is defined by the receptor binding module (RBM) in the protein segment (437-508).

2) The Kappa variant (L452R, E484Q) showed a more stable complex with a decrease in total energy (15.2KJ/mol). The authors discussed the role of E484Q mutation in stabilizing the complex but the role of L452 mutation is not discussed in detail. Does the mutation to a charged residue change the contact of L452 residue. How does it affect the vdw interaction of L452 with the corresponding residue in ACE2.

Authors: Thanks for the opportunity to comment on the L452R in the Kappa variant. A more detailed analysis, regarding the intrachain RBD contacts, highlights the effect of this mutation. In the WT RBD, the L452 makes 6 high-frequency intrachain contacts with residues 349, 350, 351, 492 and 493 (see **Supplementary Table 7**). In the Kappa variant, the fluctuation of those

contacts is reduced making the RBD more stable. From the energetic characterization, this mutation L452R does not establish any contact with the ACE2 receptor. Indeed, our energetic characterization supports this observation (see **Fig. 4c,d**). The flat profile of total and VdW energies for the WT RBD/ACE2 interface energy and VoCs is consistent with the absence of contacts mediated by L452R mutation.

3) Residue E484 in RBD is close to residue K31 in ACE2. Therefore, one would expect the mutation E484K to have a negative effect on the binding energy in the complex. However, this change is not dramatic as shown in figure 4g. Could the authors explain the reason behind this as this mutant is mostly important for being an antibody escape mutation.

Authors: The underlying mechanism that explains such a not dramatic energy increase is due to the switching of several salt bridges. In case of the WT, the E484 forms a stabilizing interaction with K31 in ACE2 denoted by a salt bridge and few strong HBs as shown in **Supplementary Table 1**. Such interaction is not disrupted in the Alpha variant by mutation N501Y; instead a stabilizing hydrophobic contact with Y41 in ACE2 is reformed. For Beta, Gamma and Kappa variants, this salt bridge is lost under mutation of residue 484 (**Supplementary Table 2-5**), however a new salt bridge (K484-E75) is formed (see **Supplementary Table 6**) in Beta and Gamma variant, which compensates the loss of primary salt bridge (E484-K31) and then offers an additional electrostatic stabilization at the RBD/ACE2 interface.

4) K417 mutation is thought to have a high impact on the binding, since it disrupts the salt-bridge with D30 on ACE2. Is this the main reason for higher energy of Beta and Gamma complexes relative to other mutations? What is the relative effect of mutation in K417 with respect to for example residue N501. A comparison between mutation in different residues would help to understand their relative importance.

Authors: In **Supplementary Table 6**, we show the list of contacts K417 forms with ACE2. In the WT, the salt bridge K417-D30 is part of the high frequency set (freq ~ 0.7). The same contact is more persistent in Alpha variant (with freq ~ 0.9) and under mutation K417N and K417T mutation in Beta and Gamma respectively, the contact frequency drops below 0.4, affecting dramatically the electrostatic contribution to the interfacial energy. For the Kappa, the salt bridge is reestablished again with a freq ~ 0.85. One can see the relative effect in terms of new contacts formed by each mutation with respect to WT. For instance, N501Y mutation is more dynamic and creates more than one contact in VoCs that are not present in the WT form, while mutations in residue 417 carry out an opposite mechanism. These mutations turn off the salt bridge contact in variants. As a summary the mutation N501Y in the Alpha, Beta and Gamma variant seems to strengthen the frequency of the Y501-Y41 contact.

5) How do these mutations affect the structure of the complex in terms of H-bonds and salt-bridges? A more structural picture of the complex is missing in the manuscript. What hydrophobic contacts are present in the WT and mutant complexes and how does the

mutations affect different type of interactions. For instance, N501 in WT is in contact with residue Y41 on ACE2. Mutation N501Y can thus have a pi-pi stacking.

Authors: Thanks for this observation. We have improved the description of the dynamics role of contacts found in our study as recommended by the reviewer. For the WT the residue 501 forms a high frequency contact (freq ~ 0.8) in ACE2, denoted by N501-Y41. The same residue in Alpha, Beta and Gamma variants change this contact into a specific hydrophobic interaction due to the attractive pi-pi interaction with a higher frequency, 0.94, 0.97, 0.98 respectively, which is a sign of the interface stabilization, as confirmed by VdW energy (see **Fig. 4j**). This contact is present in the Kappa variant but its frequency is lower than 0.7. (**Supplementary Table 6**).

6) Total energy and total LJ energy of the system are not descriptive in terms of the effect of mutations on the binding free energy. The authors should not use this simple metric to represent interaction as it is not related to the thermodynamic quantity driving stability, i.e. free energy. It is standard and accepted that free energy calculation methods such as end-point MMPBSA/MMGBSA should be used to determine the binding free energies in protein-protein complexes. The authors should use more complex methods for binding free energy computation.

Authors: As we mentioned, the energetic characterization of the RBD/ACE2 complex gives a direct correlation with structural changes in terms of the dynamic appearance/disappearance of contacts at the interface. Such analysis employs an accurate contact map (CM) determination (i.e. rCSU+OV CM). We agree with the reviewer about the use of free energy methodologies such as end-point MMPBSA/MMGBSA for the calculation of the binding free energy (ΔG_0). These methodologies have succeeded in computing the ΔG_0 of small protein-ligand complexes, and they effectively balance computational cost against accuracy. In this regard, the scientific community has already reported the ΔG_0 for the RBD/ACE2 complex using those approaches. For instance, the WT binding energy varies in the range of -50 to -100 kcal/mol^{6,7,14-17}. For the Alpha variant, the ΔG_0 varies in the range of -50 to -90 kcal/mol^{14,18-20}. For the Gamma and Kappa variants, ΔG_0 gives about -65 kcal/mol and -70 kcal/mol respectively^{15,18}. The large range of ΔG_0 values reported in those computational studies for the RBD/ACE2 complex in WT and VoCs is a clear indication of a convergence issue for large protein-protein complexes. At this moment, we are aware that an accurate ΔG_0 calculation will require large computational resources beyond the capabilities of our studies.

7) Fig4 the y-label for both left and right columns are the same.

Authors: Thanks for noticing it and we have corrected it in the revised MS.

References

- 1 Paul, L. A., Daneman, N., Brown, K. A., Johnson, J., van Ingen, T., Joh, E., . . . Buchan, S. A. Characteristics Associated With Household Transmission of Severe Acute Respiratory Syndrome Coronavirus 2 (SARS-CoV-2) in Ontario, Canada: A Cohort Study. *Clin. Infect. Dis.*, doi:10.1093/cid/ciab186 (2021).
- 2 Campbell, F., Archer, B., Laurenson-Schafer, H., Jinnai, Y., Konings, F., Batra, N., . . . le Polain de Waroux, O. Increased transmissibility and global spread of SARS-CoV-2 variants of concern as at June 2021. *Eurosurveillance* **26**, 2100509 (2021).
- 3 Khateeb, J., Li, Y. & Zhang, H. Emerging SARS-CoV-2 variants of concern and potential intervention approaches. *Critical Care* **25**, 244, doi:10.1186/s13054-021-03662-x (2021).
- 4 Tees, D. F., Woodward, J. T. & Hammer, D. A. Reliability theory for receptor–ligand bond dissociation. *The Journal of Chemical Physics* **114**, 7483-7496 (2001).
- 5 Poma, A. B., Chwastyk, M. & Cieplak, M. Polysaccharide-Protein Complexes in a Coarse-Grained Model. *J Phys Chem B* **119**, 12028-12041, doi:10.1021/acs.jpcc.5b06141 (2015).
- 6 Khan, A., Gui, J., Ahmad, W., Haq, I., Shahid, M., Khan, A. A., . . . Mohammad, A. The SARS-CoV-2 B.1.618 variant slightly alters the spike RBD–ACE2 binding affinity and is an antibody escaping variant: a computational structural perspective. *RSC Advances* **11**, 30132-30147, doi:10.1039/D1RA04694B (2021).
- 7 Williams, A. H. & Zhan, C.-G. Fast Prediction of Binding Affinities of the SARS-CoV-2 Spike Protein Mutant N501Y (UK Variant) with ACE2 and Miniprotein Drug Candidates. *The Journal of Physical Chemistry B* **125**, 4330-4336, doi:10.1021/acs.jpcc.1c00869 (2021).
- 8 Gobeil, S. M., Janowska, K., McDowell, S., Mansouri, K., Parks, R., Manne, K., . . . Acharya, P. D614G Mutation Alters SARS-CoV-2 Spike Conformation and Enhances Protease Cleavage at the S1/S2 Junction. *Cell Rep* **34**, 108630, doi:10.1016/j.celrep.2020.108630 (2021).
- 9 Mansbach, R. A., Chakraborty, S., Nguyen, K., Montefiori, D. C., Korber, B. & Gnanakaran, S. The SARS-CoV-2 Spike variant D614G favors an open conformational state. *Sci Adv* **7**, doi:10.1126/sciadv.abf3671 (2021).
- 10 Gobeil, S. M., Janowska, K., McDowell, S., Mansouri, K., Parks, R., Stalls, V., . . . Acharya, P. Effect of natural mutations of SARS-CoV-2 on spike structure, conformation, and antigenicity. *Science* **373**, doi:10.1126/science.abi6226 (2021).
- 11 Shi, R., Shan, C., Duan, X., Chen, Z., Liu, P., Song, J., . . . Yan, J. A human neutralizing antibody targets the receptor-binding site of SARS-CoV-2. *Nature* **584**, 120-124, doi:10.1038/s41586-020-2381-y (2020).
- 12 Yang, J., Petitjean, S. J. L., Koehler, M., Zhang, Q., Dumitru, A. C., Chen, W., . . . Alsteens, D. Molecular interaction and inhibition of SARS-CoV-2 binding to the ACE2 receptor. *Nature communications* **11**, 1-10 (2020).
- 13 Alsteens, D., Pfreundschuh, M., Zhang, C., Spoerri, P. M., Coughlin, S. R., Kobilka, B. K. & Müller, D. J. Imaging G protein-coupled receptors while quantifying their ligand-binding free-energy landscape. *Nat. Methods* **12**, 845-851 (2015).
- 14 Ali, F., Kasry, A. & Amin, M. The new SARS-CoV-2 strain shows a stronger binding affinity to ACE2 due to N501Y mutant. *Medicine in Drug Discovery* **10**, 100086 (2021).
- 15 Khan, A., Wei, D.-Q., Kousar, K., Abubaker, J., Ahmad, S., Ali, J., . . . Mohammad, A. Preliminary Structural Data Revealed That the SARS-CoV-2 B.1.617 Variant's RBD Binds to ACE2 Receptor Stronger Than the Wild Type to Enhance the Infectivity. *ChemBioChem* **22**, 2641-2649 (2021).
- 16 Aljindan, R. Y., Al-Subaie, A. M., Al-Ohali, A. I., Kumar D, T., Doss C, G. P. & Kamaraj, B. Investigation of nonsynonymous mutations in the spike protein of SARS-CoV-2 and its interaction with the ACE2 receptor by molecular docking and MM/GBSA approach. *Computers in Biology and Medicine* **135**, 104654 (2021).
- 17 Ali, A. & Vijayan, R. Dynamics of the ACE2–SARS-CoV-2/SARS-CoV spike protein interface reveal unique mechanisms. *Sci. Rep.* **10**, 14214, doi:10.1038/s41598-020-71188-3 (2020).

- 18 Verma, J. & Subbarao, N. Insilico study on the effect of SARS-CoV-2 RBD hotspot mutants' interaction with ACE2 to understand the binding affinity and stability. *Virology* **561**, 107-116 (2021).
- 19 Kumar, V., Singh, J., Hasnain, S. E. & Sundar, D. Possible Link between Higher Transmissibility of Alpha, Kappa and Delta Variants of SARS-CoV-2 and Increased Structural Stability of Its Spike Protein and hACE2 Affinity. *Int. J. Mol. Sci.* **22**, 9131 (2021).
- 20 Singh, J., Samal, J., Kumar, V., Sharma, J., Agrawal, U., Ehtesham, N. Z., . . . Hasnain, S. E. Structure-Function Analyses of New SARS-CoV-2 Variants B.1.1.7, B.1.351 and B.1.1.28.1: Clinical, Diagnostic, Therapeutic and Public Health Implications. *Viruses* **13**, 439 (2021).

REVIEWER COMMENTS

Reviewer #1 (Remarks to the Author):

The authors have responded to my points in a detailed, comprehensible and convincing manner, while at the same time pointing out the limitations of individual statements in the manuscript. At the same time, they have undertaken additional measurements to substantiate their conclusions (see concern 3). Only a few minor concerns remain:

a) In their response to concern 4 they argue for double-peak and triple-peak Gaussian reflecting formation of two and three bonds in parallel: "...This can be explained on the basis that ACE2 is dimeric, implying that two S1 or RBD molecules can interact at the same time..." (p.4 lines 4-5). However, as only the monomeric form of ACE2 was used this explanation is not valid!

b) Response to concern 5:

Figure R2: I miss the specification of the loading rate.

c) Response to concern 14: I still miss an indication if a maximum distance was also specified and if so, which one.

Reviewer #2 (Remarks to the Author):

(see Attachment on the next page of the file)

Reviewer #3 (Remarks to the Author):

Modification and response to my critique is satisfactory.

Reviewer # 2(Remarks to the Author):

Authors have responded satisfactory to points 1 and 3. However, for the sake of clarity in future experiments it would be desirable to extract the data for figure 2a at a contact time located at the plateau of the insets (~600 ms).

However, I still can see problems at points 2 and 4.

In the response to my query the authors say: "*the Gamma and Kappa mutants differ significantly in their KD value compared to the WT, pointing out their increased affinity towards ACE2 binding*". However, in the manuscript they say (lines 205-207): "*Taken together, our in vitro experiments show that the VoCs that emerged generally possess better ACE2 receptors binding capacity compared with WT, both in terms of affinity and stability.*" I would say that 2 out of 4 mutants does not support the adverb "generally".

In addition, the authors say in lines 197-199: "*Collectively, these experiments lead to the following calculated equilibrium dissociation constants KD (koff/kon) in ascending order: Gamma (KD = 21 + 16 nM) < Kappa (KD = 71 + 31 nM) < Beta (KD = 80 + 49 nM) ≈ Alpha (KD = 129 + 81 nM) ≈ WT (KD = 134 + 81 nM)*". Considering the standard deviation, the KD of Kappa reaches 102 nM, which is well inside of the values of beta dispersion. I would say that this is not a "significant difference", and this does not support the adverb "generally". From the KD point of view, the only mutant showing measurable differences with WT is gamma. Summing up, I think that the clearest data showing differences between WT and mutants are in figure 2a.

Regarding the theory, I am not convinced about its direct relationship with the experiments. As I said before, there is not a quantitative interplay between the experiments and the theory. The theory is not used to estimate any particular result of the experiments, such as the equilibrium constants or the binding frequency. Moreover, the MD calculation of the energy is not used to make explicit predictions about the experimental results. Indeed, it is difficult to establish a qualitative relationship between theory and experiments. In particular, figures 4e-i do not show strong variations between WT and mutants. Figure 4j shows relevant energy values variation for alpha, beta and gamma, but Kappa remain about the same than WT. The energetic changes of the conserved residues (fig. 5) show a high variability depending on the residue. Specifically, figs. 5f and 5h show that kappa is very similar to WT, but the other mutants are quite different. In addition, in their response to point 4 the authors say: "*Based on Fig. 3b we report the change in total non-bonded energy in VoCs with respect to WT. Here, it is clear Alpha, Beta and Gamma variants seem to be affected by a severe change in the electrostatic energy rather than a change in VdW energy*". All these theoretical results are in contrast with figure R3, where only gamma show significant differences of K_D with WT.

Point-by-Point Response to the Reviewers Comments

Reviewer #1 (Remarks to the Author):

The authors have responded to my points in a detailed, comprehensible and convincing manner, while at the same time pointing out the limitations of individual statements in the manuscript. At the same time, they have undertaken additional measurements to substantiate their conclusions (see concern 3). Only a few minor concerns remain:

Authors: We would like to thank the reviewer for the useful feedback and fruitful comments in the first revision, and are happy that we could respond to them in a comprehensible and convincing manner. We found them very helpful and we feel they improved the overall quality of the paper. Below you can find a point-by-point response to address the few remaining concerns.

1) In their response to concern 4 they argue for double-peak and triple-peak Gaussian reflecting formation of two and three bonds in parallel: "...This can be explained on the basis that ACE2 is dimeric, implying that two S1 or RBD molecules can interact at the same time..." (p.4 lines 4-5). However, as only the monomeric form of ACE2 was used this explanation is not valid!

Authors: We apologize for the mistake in the previous response. The reviewer is correct, for our single-molecule experiments on the model surfaces we were using the monomeric form of ACE2. However, it cannot be excluded that two RBD molecules in close proximity on the AFM tip are interacting at the same time with one ACE2 molecule on the surface, or vice versa (two ACE2 surface molecules with one RBD molecule on the AFM tip). Nevertheless, the tip-/ and surface chemistry are well established and adjusted to only probe single-molecule interactions most of the time. This is also reflected in the histograms (Supplementary Figs. 1-5), where a second and third peak are present only at a very low probability.

2) Response to concern 5: Figure R2: I miss the specification of the loading rate.

Authors: Figure R2 is a representative figure showcasing the dependence of force distribution over a specific loading rate range. In this figure, the specific loading rate range is 1000-3499 pN/s, corresponding to a pulling speed of 1 $\mu\text{m/s}$.

3) Response to concern 14: I still miss an indication if a maximum distance was also specified and if so, which one.

Authors: With respect to the previous study on the ACE2 – RBD wildtype system of Yang et al. (Supplementary Fig. 2), we excluded binding events above 30 nm, which rarely occurred. This missing piece of information has now been included in the materials and methods section (lines 510-512).

Ref:

Yang, J., Petitjean, S.J.L., Koehler, M. et al. Molecular interaction and inhibition of SARS-CoV-2 binding to the ACE2 receptor. Nat Commun 11, 4541 (2020). <https://doi.org/10.1038/s41467-020-18319-6>

Reviewer #2 (Remarks to the Author):

Authors have responded satisfactory to points 1 and 3. However, for the sake of clarity in future experiments it would be desirable to extract the data for figure 2a at a contact time located at the plateau of the insets (~600 ms). However, I still can see problems at points 2 and 4.

Authors: We thank the reviewer for his encouraging and helpful first referee report and are pleased that we were able to satisfy his concerns regarding point 1 and 3. We understand the reviewer's point of view and agree that it would be interesting to extract the data for figure 2a at contact times, located at the plateau of inserts. This process will be kept in mind for future experiments. Please find below a point-by-point response how we addressed the two remaining problematic points.

1) In the response to my query the authors say: "the Gamma and Kappa mutants differ significantly in their KD value compared to the WT, pointing out their increased affinity towards ACE2 binding". However, in the manuscript they say (lines 205-207): "Taken together, our in vitro experiments show that the VoCs that emerged generally possess better ACE2 receptors binding capacity compared with WT, both in terms of affinity and stability." I would say that 2 out of 4 mutants does not support the adverb "generally". In addition, the authors say in lines 197-199: "Collectively, these experiments lead to the following calculated equilibrium dissociation constants KD (koff/kon) in ascending order: Gamma (KD = 21 + 16 nM) < Kappa (KD = 71 + 31 nM) < Beta (KD = 80 + 49 nM) ≈ Alpha (KD = 129 + 81 nM) ≈ WT (KD = 134 + 81 nM)". Considering the standard deviation, the KD of Kappa reaches 102 nM, which is well inside of the values of beta dispersion. I would say that this is not a "significant difference", and this does not support the adverb "generally". From the KD point of view, the only mutant showing measurable differences with WT is gamma. Summing up, I think that the clearest data showing differences between WT and mutants are in figure 2a.

Authors: We thank the reviewer for these critical comments regarding the interpretation of the K_D value and their implications in terms of affinity and stability. As clearly shown in Table R1, the Gamma VoC shows a significant difference with the wildtype, as well as the Kappa VoC to a lesser extend (P-value: 1.9E-2, 0.019). The same is true, if the mutants are compared with each other. Still, according to the NEJM (New England Journal of Medicine) statistics style applied here, P-values above 5E-2 (0.05) were considered as statistically not significant (highlighted in red), whereas P-values below/equal 5E-2 (0.05) were considered as significant and P-values below/equal 5E-1 (0.01, highlighted in yellow) were considered significant obviously (highlighted in green). Nevertheless, we fully agree that the adverb "generally" is not the correct term, and changed this part of the manuscript accordingly (line 194 to 203). In addition, we added Table R1 to the Supplementary information.

Table R1 | Comparison P-values of K_D for the interaction between ACE2 and RBD-VoCs.

WT-RBD 134 ± 81 nM	Alpha-RBD 129 ± 81 nM	Beta-RBD 80 ± 49 nM	Gamma-RBD 21 ± 16 nM	Kappa-RBD 71 ± 31 nM	
	9.1E-1	7.4E-2	1.4E-5	1.9E-2	WT-RBD 134 ± 81 nM
		1.0E-1	3.5E-5	3.4E-2	Alpha-RBD 129 ± 81 nM
			2.9E-4	6.2E-1	Beta-RBD 80 ± 49 nM
				5.8E-6	Gamma-RBD 21 ± 16 nM
					Kappa-RBD 71 ± 31 nM

NEJM-Style:
 ns: $P > 5E-2$
 * $P \leq 5E-2$
 ** $P \leq 1E-2$
 *** $P \leq 1E-3$

2) Regarding the theory, I am not convinced about its direct relationship with the experiments. As I said before, there is not a quantitative interplay between the experiments and the theory. The theory is not used to estimate any particular result of the experiments, such as the equilibrium constants or the binding frequency. Moreover, the MD calculation of the energy is not used to make explicit predictions about the experimental results. Indeed, it is difficult to establish a qualitative relationship between theory and experiments. In particular, figures 4e-i do not show strong variations between WT and mutants. Figure 4j shows relevant energy values variation for alpha, beta and gamma, but Kappa remain about the same than WT. The energetic changes of the conserved residues (fig. 5) show a high variability depending on the residue. Specifically, figs. 5f and 5h show that kappa is very similar to WT, but the other mutants are quite different. In addition, in their response to point 4 the authors say: “Based on Fig. 3b we report the change in total non-bonded energy in VoCs with respect to WT. Here, it is clear Alpha, Beta and Gamma variants seem to be affected by a severe change in the electrostatic energy rather than a change in VdW energy”. All these theoretical results are in contrast with figure R3, where only gamma show significant differences of K_D with WT.

Authors: We agree with the reviewer that the MD simulations do not mimic the experimental conditions limiting a direct comparison of the equilibrium constants or binding frequency. However, we do not agree that it is difficult to establish a qualitative relationship between the theory by the MD simulations and our AFM experiments. As noticed by the reviewer, the gamma VoC and to a lesser extent the kappa VoC have emerged as VoCs, showing a reduced K_D in comparison to the WT, as observed in our AFM analysis. MD simulation revealed, as suggested in Figure 3, two different patterns in the VoCs: Alpha, Beta and Gamma VoCs showed a significant loss in their total energy (meaning a destabilized interface from an energy point of view) together with a gain in their Lennard-Jones energy (meaning creation of new contact points). Analysis of the contact frequency revealed that for three VoCs, in comparison to the WT, totally new contact points are established with ACE2 (75-484, 38-501, 42-501). By looking closer at the energy of individual residue of these VoCs (Figure 4), we observed that the gain of energy is mainly located at residue 501, which is also highlighted in the Fig. 3e. **Together with the AFM experiments, these results suggest that the destabilization of the interface enables the creation of additional key contact points around residue 501. We could**

see this as a 'key binding hotspot', providing a rationale for the decrease in K_D observed by AFM (Fig. R1).

Figure R1 | Interface destabilization between ACE2 and either RBD-Alpha, RBD-Beta or RBD-Gamma VoCs enables additional key contact points around residue 501, leading to a 'key binding hotspot' (shown as a star).

However, for the Kappa VoC, the MD simulation shows very different behavior. While this VoC seems more similar to the WT in terms of total energy and Lennard-Jones energy, we observe a redistribution of the strength of pre-existing contact points and energies all over the interface. This is in particular observed in Figure 3d, which shows a reduction of the contact frequency for the Kappa VoC in the "binding hotspot" area (around residues 501), compensated at least in part by additional contacts in the opposite part of the interface (around residues 417 and 484), as highlighted in the new panel (f) in the revised Fig. 3. MD supports the idea of a stabilized interface through multiple weaker molecular interactions located all over the interface. **In this configuration, the gain in K_D would not come from a "binding hotspot" but from a better-balanced interface from an energy point of view (Fig. R2).**

Figure R2 | No interface destabilization between ACE2 and RBD-Kappa VoC, leading to a better-balanced interface and the establishment of additional contacts around residues 417 and 484 (shown as stars).

Figure 3 | MD simulation of the RBD—ACE2 of WT and VoCs. (..). (e, f) Zoom-in on the RBD—ACE2 region around (e) RBD residue 501 and (f) around RBD residues 484 and 417 for the WT and the 4 VoCs. Sidechains are represented by sticks. Residues 38, 42 and 355 in ACE2 contacting RBD residue 501 (e) and residues 30, 31, 34 and 75 in ACE2 contacting RBD 417 or 484, respectively, are shown. (...).

While AFM experiments provide only a global measurement, MD simulations allow a better understanding of the molecular mechanisms responsible for the stability of the interface between the RBD domain of SARS-CoV-2 and the ACE2 receptor. Out-of-equilibrium, and especially when these bonds are subject to external forces (AFM under our experimental conditions, shear forces *in vivo*, etc.), a better-balanced interface may be favorable to the bound state as it will require parallel rupture of multiple bonds, rather than serial breakage of the individual bonds.

We included the additional information and discussion into the main manuscript on lines 261-263, 277, 413-416, 420-426 and 433-439.

Reviewer #3 (Remarks to the Author):

Modification and response to my critique is satisfactory.

Authors: The authors thank the reviewer for giving a positive evaluation of the manuscript.